# Development of Pranoprofen Loaded Nanostructured Lipid Carriers to Improve Its Release and Therapeutic Efficacy in Skin Inflammatory Disorders

**DOI:** 10.3390/nano8121022

**Published:** 2018-12-07

**Authors:** María Rincón, Ana C. Calpena, María-José Fabrega, María L. Garduño-Ramírez, Marta Espina, María J. Rodríguez-Lagunas, María L. García, Guadalupe Abrego

**Affiliations:** 1Department of Pharmacy, Pharmaceutical Technology and Physical Chemistry, Faculty of Pharmacy and Food Sciences, University of Barcelona, 08028 Barcelona, Spain; mariarincon5@hotmail.com (M.R.); m.espina@ub.edu (M.E.); rdcm@ub.edu (M.L.G.); 2Institute of Nanoscience and Nanotechnology (IN2UB), University of Barcelona, 08028 Barcelona, Spain; 3Department of Biochemistry and Physiology, Faculty of Pharmacy and Food Sciences, University of Barcelona, 08028 Barcelona, Spain; mjfabrega.f@gmail.com (M.-J.F.); mjrodriguez@ub.edu (M.J.R.-L.); 4Institute of Biomedicine, University of Barcelona, 08028 Barcelona, Spain; 5Centro de Investigaciones Químicas, Universidad Autónoma del Estado de Morelos, Cuernavaca 62000, Morelos, Mexico; lgarduno@uaem.mx; 6Institut de Recerca en Nutrició i Seguretat Alimentària (INSA), Universitat de Barcelona (UB), 08921 Santa Coloma del Gramanet Barcelona, Spain; 7Department of Chemical and Instrumental Analysis, Faculty of Chemistry and Pharmacy, University of El Salvador, Ciudad Universitaria, 3026 San Salvador, El Salvador; escobar.ues@gmail.com

**Keywords:** pranoprofen, nanostructured lipid carriers, penetration enhancers, linoleic acid, skin delivery, anti-inflammatory activity

## Abstract

Pranoprofen (PF)-loaded nanostructured lipid carriers (NLCs), prepared using a high-pressure homogenization method, have been optimized and characterized to improve the biopharmaceutical profile of the drug. The optimized PF-NLCs exhibited physicochemical characteristics and morphological properties that were suitable for dermal application. Stability assays revealed good physical stability, and the release behavior of PF from these NLCs showed a sustained release pattern. Cell viability results revealed no toxicity. Ex vivo human skin permeation studies in Franz diffusion cells were performed to determine the influence of different skin penetration enhancers (pyrrolidone, decanol, octanoic acid, nonane, menthone, squalene, linoleic acid, and cineol) on skin penetration and retention of PF, being the highest dermal retention in the presence of linoleic acid. The selected formulations of NLCs exhibited a high retained amount of PF in the skin and no systemic effects. In vivo mice anti-inflammatory efficacy studies showed a significant reduction in dermal oedema. NLCs containing linoleic acid presented better anti-inflammatory efficacy by decreasing the production of interleukins in keratinocytes and monocytes. The biomechanical properties of skin revealed an occlusive effect and no hydration power. No signs of skin irritancy in vivo were detected. According to these results, dermal PF-NLCs could be an effective system for the delivery and controlled release of PF, improving its dermal retention, with reduced dermal oedema as a possible effect of this drug.

## 1. Introduction

Skin provides protection from environmental injuries, prevents microbial invasion, regulates temperature, and maintains hydration. The skins infrastructure provides the necessary components to form a protective barrier. In particular, the outer epidermal layer, the stratum corneum (SC), normally retains enough water and acts as a hydro-lipid film constituting an effective first-line of defense against the outside world molecules [1].

The cutaneous permeability barrier resides in the extracellular lipids, mainly ceramides, free fatty acids, and cholesterol, which form extracellular lipid-enriched lamellar membranes between the corneocytes that block the movement of water and the electrolytes [2]. The lipid layers are anisotropic, meaning that the layers permeability is direction-dependent. This property greatly affects the diffusion characteristics in the skin, since diffusion through the SC is intercellular [3,4].

Skin inflammation is one of the most common skin problems. Skin inflammation is a wide term that describes typical inflammation symptoms, such as redness, warmth, pain, and swelling. There are a broad variety of dermatological diseases that include inflammatory reactions in the skin that range in severity from a mild skin rash to severe dermatitis, arising from a systemic disease or from generalized infections. Other diseases, such as psoriasis and acne, are also considered to entail various inflammatory implications at some stages [5].

In chronic diseases, in skin affected by inflammatory disorders like atopic dermatitis or psoriasis, the differentiation process of the keratinocytes, the biosynthesis of the SC, its lipid composition, and its organization are altered [6].

Topical products have been developed to minimize the flux of the drug through the skin, whilst maximizing its retention in the skin. However, the troublesome penetration of drugs across the stratum corneum, limits the drug’s efficacy. This is the reason why we should consider the increase in drug penetration across the SC and/or epidermis when we develop topical drug delivery systems. The primary aim of dermatological therapeutic products is to increase the retention of therapeutic drugs in the skin rather than the penetration of the drug through the skin. The major barriers to the topical delivery of drugs are that they permeate poorly across the SC, and they are not easily retained in the skin for localized therapy and local irritation.

Pranoprofen (PF), or 2-(5H-[1] benzopyrano-[2,3-b]-pyridin-7-yl) propionic acid is a non-steroidal anti-inflammatory drug (NSAID). It is an amphoteric compound with carboxyl and pyridinic moieties, with a molecular weight of 255.27 g/mol. It has poor aqueous solubility as a function of the media pH [7,8]. The main effect of this drug is the inhibition of cyclooxygenase, an enzyme of the arachidonic acid cascade that generates mediators such as thromboxanes and prostaglandins involved in some aspects of the inflammatory response [9]. Despite the high anti-inflammatory and analgesic potency of this drug, as well as the minimal risk of side effects on the gastrointestinal tract, the oral administration of PF is somehow limited because of its inadequate biopharmaceutical profile. Indeed, PF has poor aqueous solubility (although it is freely soluble in most organic solvents) and it has a short plasmatic half-life [9,10,11].

To improve the effects of the drug and diminish its side effects, it is necessary to modulate its permeability. For this purpose, penetration enhancers can be used in a formulation for improving the transdermal drug delivery and also for the enhancers ability to reversibly decrease the barrier resistance. Therefore, penetration enhancers could be used to enhance the drug retention in the skin [12]. These penetration enhancers must have certain properties, i.e., they should be pharmacologically inert; they should not be toxic, irritating or allergenic; on application, the onset of action should be immediate, and the duration of the effect should be predictable; and they must be chemically and physically suitable for the formulation, active ingredients, and excipients.

Additionally, linoleic acid is an essential fatty acid and an important building block for the intercellular lipid complex. A deficiency in linoleic acid leads to altered barriers, and it has been demonstrated that patients with atopic dermatitis have decreased linoleic acid metabolites in the SC [1].

Nanocarriers applied to the skin may alter the flux, provide drug deposition and localization, and even selectively permeabilize the SC. Thus, these nanosystems may enable the consecutive and long-term administration of lipophilic drugs, targeting only the areas of the disease and improve the drugs pharmacokinetics by optimizing the dose for treatment [13].

It has been identified that NLCs could improve some limitations of solid lipid nanoparticles (SLNs), such as the relatively low drug payload and drug expulsion during storage. Unlike the crystalline structure of SLNs, NLCs elicit imperfections in the lipid core matrix, which allows for a higher payload with less drug expulsion [13].

NLCs ensure close contact to the SC owing to its unique lipid composition and smaller particle size; thus, NLCs are known for excellent adhesion, skin hydration, and occlusion properties, thereby enhancing dermal drug delivery [5]. The combination of the occlusive effect of NLCs and the SC lipid disturbance attributed to the chemical enhancers arises as an appealing strategy to overcome SC, resulting in transdermal delivery [14].

The main goal of this study was the design and development of a new delivery system for PF-loaded NLCs, for the dermal administration of PF, prepared using a high-pressure homogenization technique. After selecting the critical formulation variables that affect the physicochemical properties of the NLCs and taking into account that a factorial design provides maximum information from the fewest experiments, a 2^3^+star central composite factorial design was then employed to plan and perform the experiments [15]. To define how the presence of different skin penetration enhancers (pyrrolidone, decanol, octanoic acid, nonane, menthone, squalene, linoleic acid, and cineol) affects the PF retention in human skin in an ex vivo experiment, the selected NLCs were characterized for in terms of morphometry, spreadability, rheological behavior, and physically stability. Possible interactions between the drug and the lipid matrix were studied. In vitro release profile, cytotoxicity studies, ex vivo skin permeation of formulations, histological characterization, in vivo transepidermal water loss, as well as skin tolerance, and the anti-inflammatory efficacy of PF-NLCs developed were also assayed.

## 2. Materials and Methods

### 2.1. Materials

PF was supplied by Alcon Cusi (Barcelona, Spain); LAS (PEG-8 Caprylic/Capric Glycerides) and PAT (Precirol^®^ ATO 5 (glycerol mono, di and tripalmitostearate)) were gifted by Gattefossé (Gennevilliers, France); and Castor oil and Tween^®^ 80 were purchased from Sigma-Aldrich. The penetration enhancers, linoleic acid (9-cis,12-cis-linoleic acid), decanol (decyl alcohol), menthone ((2S,2R)-2-isopropyl-5-methylcyclohexanone), nonane (n-nonane), pyrrolidone (*N*-Dodecylpyrrolidone), octanoic acid, squalene (2,6,10,15,19,23-hexametyltetracosane), and cineol were acquired from Sigma Aldrich (Madrid, Spain).

The HaCat was provided by Cell Lines Service (CLS, Eppelheim, Germany) and the 3-(4,5-dimethylthiazol-2-yl)-2,5-diphenyltetrazolium bromide (MTT) used for cell viability was acquired from Invitrogen Alfagene^®^ (Carcavelos, Portugal). The purified water used in all the experiments was obtained from a MilliQ Plus System (Millipore Corporation, Bedford, MA, USA). All other chemicals and reagents used in the study were of analytical or HPLC-grade (Fischer Chemical, Loughborough, UK).

### 2.2. Methods

#### 2.2.1. Preparation of NLCs

The NLCs were obtained by a high pressure homogenization method as described previously [16]. Briefly, the lipid phase (5 wt % with regard to the total formulation) containing LAS: Castor oil (ratio (75/25) was kept constant) (LAS-CO) as liquid lipid (LL) and Precirol^®^ ATO 5 (PAT) as solid lipid (SL) (also in selected formulations 5% of acid linoleic) was melted in a water bath at 85 °C to obtain a homogeneous lipid solution. An aqueous solution with Tween^®^ 80 heated at the same temperature was added to the hot lipid phase, and a pre-emulsion was generated using an Ultra-Turrax T25 (IKA, Staufen, Germany) at 8000 rpm for 45 s, which was passed through a high-pressure homogenizer (Homogeniser FPG 12800, Stansted, UK). The production conditions were 85 °C and three homogenization cycles (Appendix A). The nanoemulsion obtained was subsequently cooled down to room temperature (RT), recrystallizing the lipid, and forming the NLCs.

#### 2.2.2. Design of Experiments

A 2^3^+star central composite factorial design was applied to optimize the formulation parameters as described previously [11]. Three independent variables (concentration of PF (cPF), concentration of solid lipid regarding liquid lipid (cSL/L), and concentration of Tween^®^ 80 (cTW)), and four dependent variables (mean particle size (Z-Ave), polydispersity index (PI), zeta potential (ZP), and entrapment efficiency (EE)) were studied. The variables were analyzed at five different levels coded as −α, −1, 0, 1, and +α. The value of alpha (1.68) was calculated to meet the design rotatability (Appendix A).

According to the composite central design matrix, generated by Statgraphics^®^ Centurion XVI Software, a total of 16 experiments (eight factorial points, six axial points, and two replicated center points) are summarized in Appendix A.

#### 2.2.3. Physicochemical Characterization

##### Particle Size and Zeta Potential

Z-Ave and PI of PF-NLCs were determined by photon correlation spectroscopy (PCS) using a Zetasizer Nano ZS (Malvern Instruments, Malvern, UK) [17]. For these measurements, samples were diluted (1:20) with Milli-Q water and were carried out in triplicate in 10 mm diameter cells at 25 °C. In addition, using this instrument, the ZP was determined using the Electrophoretic Light Scattering (ELS) technique. Additionally, particle size analysis of optimized PF-NLCs were performed by laser diffraction (LD) [15,18] using a Malvert Mastersizer 2000 (Malvert Instrument, UK) in order to evaluate the presence of large particles. Data were evaluated using the volume distribution method, and it was applied the Mie analysis determining d(0.1), d(0.5), and d(0.9). 

##### Entrapment Efficiency

EE was indirectly determined measuring the concentration of the non-entrapped drug in the dispersion medium using a reversed phase high-performance liquid chromatography (RP-HPLC) [19,20]. Briefly, each sample was diluted with PBS pH 7.4 (1:20) and was separated using a filtration/centrifugation technique with Ultracel YM-100 (Amicon^®^ Millipore Corporation, Bedford, MA, USA) centrifugal filter devices at 6000 rpm for 30 min (Sigma 301K 8 centrifuge, Osterode am Harz, Germany), as described by Gonzalez et al. [21]. Validation of the developed methodology was performed in accordance to international guidelines (EMEA, 2011), including the evaluation of linearity, sensitivity, accuracy, and precision. The EE was calculated using the following equation:
EE (%) = ((Total amount of PF − Free PF)/Total amount of PF)(1)

The system consisted of a Waters 1525 pump (Waters, Milford, CT, USA) with a UV-Vis 2487 detector (Waters, Milford, CT, USA) (λ = 235 nm) and Kromasil C-18, 150 × 4.6 mm, 5 µm column at a flow rate of 1 mL/min. The mobile phase was methanol/glacial acetic acid 5% (70:30; v:v).

##### Transmission Electron Microscopy

The morphological examination of the optimized NLCs was performed by transmission electron microscopy (TEM) using a JEOL 1010 Instrument (Jeol Inc., Peabody, MA, USA), with a 60,000× magnification. A drop of the sample (without previous dilution) was added onto a copper grid coated with carbon film and negative stained with uranyl acetate solution 2%. The grids were dried around 24 h for later reading [22].

##### X-ray Spectroscopy (XRD)

Samples (centrifuged and dried optimized PF-NLCs and solid components) were sandwiched between 3.6 µm films of polyester and exposed to Cu-Kα radiation (45 kV, 40 mA, λ = 1.5418 Å) in the range (2θ) from 2° to 60° with a step size of 0.026° and a measuring time of 196 s per step [11].

##### Fourier transform infrared (FTIR)

FTIR spectra of the centrifuged and dried optimized PF-NLCs and solid components were obtained using a Thermo Scientific Nicolet iZ10 with an ATR diamond and a DTGS detector. The scanning range was 525–4000 cm^−1^ [23].

##### Extensibility (Spreadability)

A weight of 0.05 g of formulation was placed within a circle of 10 cm diameter pre-marked on a glass plate, over which a second glass plate was placed, as centered as possible. Increasing standard weight pieces (5, 10, 15, 25, and 50 g) were replaced and allowed to rest on the upper glass plate for 1 min. The increase in the diameter due to the formulation spreading was noted. Each formulation was tested in triplicate at RT. The formulations were analyzed in accordance with the best kinetic model.

##### Rheology

Rheological properties were analyzed at 25 °C ± 2 °C using a Haake Rheo Stress 1 rheometer (Thermo Fisher Scientific, Karlsruhe, Germany) equipped with a cone rotor C60/2-Ti (60 mm diameter, 2° angle, 0.105 mm gap between cone-plate). For the measurements, the device was connected to a thermostatic circulator Thermo Haake Phoenix II + Haake C25P and a computer provided with the Haake Rheowin^®^ Job Manager v 3.3 software (Thermo Electron Corporation, Karlsruhe, Germany) to execute the tests and Haake Rheowin^®^ Data manager v 3.3 software (Thermo Electron Corporation, Karlsruhe, Germany) to perform the analyses of the obtained data, as explained previously [24]. Viscosity curves and flow curves were recorded under rotational runs at 25 °C for 3 min during the ramp-up period from 0 to 50, 1 min at 50 (constant share rate period), and finally 3 min during the ramp-down period from 50 to 0. Viscosity values at 50 s^−1^ were determined after 3 days of the production. All measurements were performed in triplicate.

#### 2.2.4. Stability

Short physical stability was assessed after 3, 8, 15, and 30 days analyzing light backscattering (BS) profiles by using the Turbiscan^®^Lab (Formulaction Co., L’Union, France). For this purpose, the optimized PF-NLCs were stored at 4, 25, and 37 °C for one month, and a cylindrical glass measurement cell was filled with 20 mL of sample. The radiation source was a pulse near infrared light (λ = 880 nm) and it was received by transmission and backscattering detectors at an angle of 90 and 45° from the incident beam, respectively. It could be considered that if the BS profiles had a deviation of ≤2%, there were no significant variations on particle size and variations more than ±10% indicated unstable formulations [25].

For the study of long-term stability, the storage stability was assayed at 4, 25, and 37 °C for six months. The Z-Ave, PI, ZP, and EE were monitored for 24 h and after 3, 8, 15, 30, 60, 90, and 180 days to verify if potential changes occurred [26].

#### 2.2.5. *In Vitro* Studies

##### Release

Amber glass vertical Franz diffusion cells (FDC 400; Crown Glass, Somerville, NJ, USA) with dialysis membranes (Dialysis Tubing Visking, Medicell International Ltd., London, UK) were used to address the PF release studies. The membranes were previously hydrated in methanol-water (7:3; *v*/*v*) for 24 h before being mounted in the Franz diffusion cell. Phosphate buffer saline (PBS) at pH 7.4 kept at 32 ± 0.5 °C was used as a receptor, which was under continuous stirring at 700 rpm assuring sink conditions. Samples of 1440 µL of NLCs-F3 and NLCs-F3-L and 2160 µL of NLCs-F3 and NLCs-F3-L formulations were placed in the donor compartment in direct contact with the membrane (0.64 cm^2^), then 300 µL of sample were collected with a syringe from the receptor compartment at predefined times and the volume withdrawn was replaced by an equivalent volume of fresh PBS pH 7.4 at the same temperature [27]. Samples were analyzed by RP-HPLC as described previously for the EE.

The amount of PF released was adjusted to four kinetic models: Zero order, first order, Korsmeyer-Peppas, and Weibull functions by nonlinear least-squares regression using the WinNonLin^®^software (WinNonlin^®^Professional edition version 3.3; Pharsight Corporation, Sunnyvale, CA, USA) and Graphpad prism version 6 Demo. The model fitting appropriateness was determined by calculation of the coefficient of determination (r^2^), and a discrimination models parameter, as well as the Akaike’s information criterion (AIC), which is a measure of the best fit based on the maximum likelihood and it was calculated by the equation:AIC = n × ln (WSSR) + 2 × p(2)
where n is the number of dissolution data points, p is the number of the parameters of the model, and WSSR is the weighed sum of square of residues as described in Reference [22].

##### Cell Viability Assay

*In vitro* MTT cytotoxicity assay was performed in a human model of epidermis using the cell line HaCaT (human keratinocytes) from ATCC as in Reference [28]. For that, 2 × 10^5^ cells/mL cells were cultured in 96-well plates (Costar, Fisher Scientific, Madrid, España) for 24 h in Dulbecco’s Modified Eagle Medium (DMEM High Glucose) (Life Technologies, Delhi, India) supplemented with 10% of fetal bovine serum (FBS), 25 mM HEPES, 1% non-essential amino acids, penicillin (100 U/mL), and streptomycin (100 µg/mL) (Gibco BRL, Gaithersburg, MD, USA). Cells were incubated at 37 °C in a 5% of CO_2_ atmosphere. Then they were incubated with different concentrations of NLCs-F3-L and NLCs-F9-L using a stock PF concentration of 0.712 mg/mL and 0.470 mg/mL, respectively. Dilutions assayed were 1/2, 1/5, 1/10, 1/20, 1/50, 1/100, 1/500, 1/1000, and 1/2000. After 24 h of incubation, cells were washed with sterile PBS 1x and incubated with MTT solution for 2 h. Then, the solution was removed and the cells were lysed with DMSO 98% to release the purple crystal formed by intracellular dehydrogenases of viable living cells. The absorbance was read using a Microplate Autoreader at an excitation/emission of 540/630 nm (Modulus Microplate Multicode Reader-Turner Biosystems, Sunnyvale, CA, USA). Two negative controls were also tested: a negative control of cells without any treatment and a blank control of cells treated with a blank solution used to solve nanoparticles. Both were processed for comparison. Absorbance values were considered proportional to cell viability and the percentage cell viability was calculated accordingly as: (%) viable cells = (ABS treated cells/ABS control cells) × 100.

Statistical analysis was performed using Graphpad Prism^®^ software 5 Inc, CA USA. Results were expressed as the mean ± SEM (standard error) of at least three independent experiments. Differences between groups were tested by One-Way ANOVA. Tukey’s test was used to compare the means of the ranks among groups. Results were considered statistically significant at *p* < 0.05.

#### 2.2.6. Ex Vivo Studies

##### Human Skin Permeation Assay

For the selection of penetration enhancers (pyrrolidone, decanol, octanoic acid, nonane, menthone, squalene, linoleic acid, and cineol), skin permeation studies were performed using Franz diffusion cells (FDC 400; Crown Glass, Somerville, NJ, USA) that used human skin membranes and PBS at pH 7.4 as a medium with a skin area available for permeation of 0.64 cm^2^. Samples of 1 mL of PF diluting in PBS pH 7.4 (1 mg/mL), with 5% *v*/*v* of each penetration enhancer, were placed in the donor compartment. Samples (300 μL) were withdrawn from the receptor compartment at fixed times and replaced by an equivalent volume of PBS pH 7.4 solution at the same temperature. At the end of the study, the skin was used to determine the amount of drug retained. The skin was cleaned using a 0.05% solution of sodium laurylsulphate and washed in distilled water. The diffusional area of the skin in direct contact with the formulation was isolated, weighed, and treated with methanol: water (50:50, v:v) for 20 min under sonication in an ultrasound bath.

Ex vivo permeation experiments were carried out using human skin obtained from the abdominal region of a 38-year old healthy woman during plastic surgery (Barcelona-SCIAS Hospital, Barcelona, Spain). The patient provided written informed consent. The experimental protocol was approved by the Bioethics Committee of the Barcelona-SCIAS Hospital (reference number: BEC/001/16). All samples were made using skin from the same donor to diminish the variability of the response due to biological differences. The skin was cut starting from the SC with a thickness of 0.4 mm pieces using a dermatome (Model GA 630, Aesculap, Tuttlingen, Germany), and storage at −20 °C [29]. The skin permeation studies were performed in Franz diffusion cells (FDC 400; Crown Glass, Somerville, NJ, USA) using human skin membranes and PBS pH 7.4 as the receptor medium allowing sink conditions. The skin area available for permeation was 0.64 cm^2^. Samples were placed in the donor compartment (covered with parafilm^®^ to prevent it from leaking), in contact with the SC. Samples (300 μL) were withdrawn from the receptor compartment at fixed times and replaced by an equivalent volume of PBS pH 7.4 solution at the same temperature. At the end of the study, the skin was used to determine the amount of drug retained. The skin was cleaned using a 0.05% solution of sodium laurylsulphate and it was washed in distilled water. The diffusional area of the skin in direct contact with the formulation for the drug extraction was isolated, weighed, and treated with methanol: water (50:50, v:v) under sonication for 20 min using an ultrasound bath. The amount of PF permeated and retained in the skin was determined by HPLC. Experimental data were processed using the Graphpad prism software (version 5.0) and Laplace software (Micromath. Inc., Salt Lake City, UT, USA), according to the infinite dose method (OECD, 2000). The steady-state flux across the skin J (µg/cm^2^s) and the transdermal permeability coefficient (cm/s) were obtained by applying the following equations:P_1_ = K × L(3)
P_2_ = D/L2(4)
Q = P_1_ × A × C_0_/S × (S/P_2_)^1/2^ × sin (S/P_2_)^1/2^(5)
Kp = P_1_ × P_2_(6)
J = Kp × C_0_(7)
where K is the membrane/donor phase partition coefficient, L is the effective length of diffusion through the skin, and D is the diffusion coefficient. Q is the drug permeated, A, C_0_, and S are the membrane area, concentration of the PF at time zero in the donor compartment, and the Laplace operator, respectively. Considering the pharmacokinetic parameters of the PF for young and elderly subjects, the predicted steady-state plasma concentration of the drug that would penetrate the skin after topical application was obtained using the following equation:Css = (J × A)/CLp(8)
where Css is the plasma steady-state concentration, J is the flux determined in this study, A is the hypothetical area of application (in this case, 100 cm^2^), and the plasmatic clearance (1146.60 cm^3^h and 609.00 cm^3^h, respectively, for young and elderly subjects as outlined in Reference [8].

##### Histological Analysis

Ears of the mice were subjected to the topical application of arachidonic acid (AA) for 5 min as a model of skin inflammation. Then PF (0.75 mg/mL), NLCs-F3-L or NLCs-F9-L formulations were applied for 1 h to study the anti-inflammatory effect. Ears of the non-treated animals were used as the control condition. After that, the mice were euthanized, and the ears were immediately excised, rinsed with PBS pH 7.4, and set overnight in 4% buffered formaldehyde at room temperature, and then the samples were embedded in paraffin wax. Transversal sections (5 µm) were stained with hematoxylin and eosin and were finally viewed on blind coded samples under a light microscope (Olympus BX41 and camera Olympus XC50) for the evaluation of the ear inflammation.

#### 2.2.7. In Vivo Assays

##### Skin Integrity Parameters

In vivo assays were approved by the Ethics Committee of the University of Barcelona (IRB00003099), and they followed the recommendations of the Declaration of Helsinki [30]. Ten volunteers in the age range of 25–60 years were recruited after medical screening, after they were informed of the nature of the study and of the procedures involved; whereupon the individuals gave written informed consent. The individuals were allowed to stay in the test room for at least 30 min prior to the measurements (room conditions 22 ± 2 °C and 40–50% relative humidity) and they were warned not use skin-care cosmetics on the flexor side of the left forearm the day before the study. Circles of 7 cm in diameter were made with a special marker of the skin. Samples of 50 µL, equivalent to 37.5 µg of NLCs-F3 or NLCs-F3-L, and 25 µg of NLCs-F9 or NLCs-F9-L, were deposited with an automatic micropipette in the middle of the circle by means of a soft circular movement with the thumb to facilitate the distribution of the formulation and it was measured with a chronometer (2 turns/second). A total of 20 circles in a clockwise direction were performed. Measurements were performed before applying the selected PF-NLCs at 5 and 60 min after application. 

The measurement of the quantity of water that passed through the epidermal layer of the skin to the surrounding atmosphere by diffusion and evaporation processes was performed by the transepidermal water loss (TEWL) assessment using a DermaLab^®^ module (Cortex Technology, Hadsund, Denmark) [31]. To measure the TEWL, the probe, a small hollow cylinder, was held on the skin surface for 2 min. TEWL values (g/h) were reported as the mean of 10 replications ± SD.

The measurements of stratum hydration (SCH) were carried out using a Corneometer^®^ 825, which was mounted on a Multi Probe Adapter^®^ MPA5 (Courage & Khazaka Electronics GmbH, Cologne, Germany). The measurement was performed by the capacitance method that used the relatively high dielectric constant of water compared to the ones of other substances of the skin. Values (arbitrary units, AU) were reported as the mean of 10 replications ± SD.

##### Anti-Inflammatory Efficacy

Model of Mice Ear Inflammation Induced with TPA

Anti-inflammatory activity of the selected NLCs was evaluated using a topical inflammation model using TPA (12-*O*-tetradecanoylphorbol 13-acetate) to induce mouse ear inflammation [8]. It was used for each of the formulation groups of six adult male Wistar CD-1 mice, with a weight ranging from 20 to 25 g. Edema was induced by the topical application of 2.5 μg per ear of TPA dissolved in ethanol. Formulations such as TPA were applied (50 μL/ear) to both sides of the right ear. The ear swelling was measured before TPA application and 4 h after, and the edema was expressed as the increase in thickness.

##### Transcriptional Analysis of Key Inflammatory Biomarkers

The inflammatory process associated with NLCs-F3-L and NLCs-F9-L treatment, in an in vivo ear mice model of inflammation, induced by arachidonic acid 0.5% was evaluated by RT-qPCR. Mice ears were treated with 50 µL of arachidonic acid 0.5% for 15 min to induce the inflammation, and then the ears were treated with 0.5 mL of blank solution, NLCs-F3-L or NLCs-F9-L. Two controls were established for comparison purposes: a negative control animal that did not receive any intervention, and a sham ( positive control) for whom inflammation was induced with arachidonic acid, but did not receive the anti-inflammatory treatment. Gene expression level changes of interleukin-6 (IL-6), interleukin-1β (IL-1β), tumor necrosis factor-α (TNF-α), and interleukin-10 (IL-10), involved in cellular inflammation were analyzed in small sections of the mice ear tissue after optimized NLCs-L treatment. Total RNA was extracted after tissue homogenization in cold TRI Reagent^®^ (Sigma Aldrich) and the concentration and purity were checked in a Thermo Scientific Nano Drop TM 2000 Spectrophotometer. RNA integrity was evaluated by visualization of 28S and 18S rRNAs in agarose/formaldehyde gel electrophoresis. cDNA was transcribed from RNA using the High Capacity cDNA Reverse Transcription Kit (Applied Biosystem^TM^, Fisher Scientific) following the manufacturer’s recommendations. Real time PCR was performed in a SetpOne Plus PCR cycler (Applied Biosystems) using the SYBR^®^ Green PCR Master Mix (Applied Biosystems) and specific oligonucleotides for genes were analyzed and GAPDH as the housekeeping gene. Statistical analysis was performed using the Graphpad Prism^®^ software 5 Inc, CA USA. Results were expressed as the mean ± SEM (standard error) of at least three independent experiments. Differences between groups were tested by One-Way ANOVA. Tukey’s test was used to compare the means of the ranks among groups. Results were considered statistically significant at *p* < 0.05.

##### Skin Irritation Testing

Skin irritation potential of the selected NLCs was assessed using the Draize skin irritation test on New Zealand male albino rabbits (weighing 2.5–3 Kg; *n* = 3). This study was approved by the animal research ethical committee of the University of Barcelona according to the regulations of the Spanish Government (Law 32/2007 of 7 November 2007, and Royal Decree 1201/2005 of 10 October). The rabbits were acclimatized for 5 days before the study, and the rabbits back hair was shaved 12 h before the assay. Selected NLCs were assayed using 0.9% (*w*/*v*) NaCl solution as the Control, where 0.5 mL of each formulation was applied on the hair-free rabbit’s back by uniform spreading within the area of 5 cm^2^ and it was left uncovered. The animals were examined at 24, 48, and 72 h after the application of the formulations. The sites of contact were inspected for dermal reactions such as erythema and edema. Taking into account the primary irritation index value, the formulations were classified as “non-irritant” (<0.5), “irritant” (2–5), or “highly irritant” (5–8) as described in References [32,33].

## 3. Results

### 3.1. Design of Experiments and Physicochemical Characterization

A 2^3^+star central composite factorial design was used to optimize the formulation parameters to obtain suitable PF-NLCs samples for dermal administration. The effect of the independent variables on the physicochemical properties of the NPs was studied (Appendix A).

The Z-Ave values of the 16 formulations varied from 204.57 ± 3.21 to 452.53 ± 4.25 nm. Another important physicochemical property of the NLCs was the PI. The PI values of the 16 formulations varied from 0.188 ± 0.011 to 0.603 ± 0.023.

Values obtained in Appendix A showed that all the formulations obtained by the factorial design had a net negative charge with ZP values of −7.88 mV to −11.10 mV.

The EE values of the 16 formulations varied from 84.05% to 99.82%.

In accordance with the objective of this study and the factorial design results, NLCs-F3 and NLCs-F9 formulations showed the best Z-Ave and PI values with a percentage of incorporated PF in the lipid matrix (EE) of around 99% (Table 1).

Particle size of the optimized NLCs and the ones with 5% of linoleic acid were further confirmed by LD analysis (Appendix A).

The low Z-Ave result obtained was 0.196 d (0.5), which indicated that more than 50% (V) of the particles were smaller than 196 nm in all cases.

HPLC Validation Methodology

The HPLC method was developed and validated with the goal of determining the amount of PF. Therefore, parameters such as linearity, selectivity, sensitivity, accuracy, and precision were assessed. It was established by the measurement of five calibration curves, and these curves ranged between 12.5–250 and 0.78–12.5 µg/mL. The results of the limit of detection and the limit of quantification obtained were 1.099 ± 0.711 and 3.332 ± 2.157 µg/mL, respectively.

Considering the results obtained for the parameters of specificity, accuracy, precision, linearity, and sensitivity, the method under validation had proved to be suitable for the analysis and quantification of PF in the ranges of 12.5–250 µg/mL and 0.78–12.5 µg/mL.

#### 3.1.1. Transmission Electron Microscopy (TEM)

The size and surface morphology of the optimized NLCs were determined by TEM (Figure 1). Size results by TEM were according to those obtained by photon correlation spectroscopy (PSC); where the results showed almost spherical shapes and no particle aggregation phenomena were found.

#### 3.1.2. X-ray Spectroscopy (XRD)

XRD was used to analyze the amorphous or crystalline state of the PF incorporated NLCs. Intense and sharp peaks for PF, Precirol® ATO 5(PAT), and the solid mixture of lipids were shown (Figure 2). This indicated that these components had a crystalline structure.

#### 3.1.3. Fourier Transform Infrared (FTIR)

The FTIR of the PF-NLCs showed that no new covalent bonds were made between the lipid matrix and the PF. The characteristic peak of the PF and PAT were exhibited (Figure 3). The optimized NLCs profile was the same as the blank NLCs indicating that there were only weak interactions between the components.

#### 3.1.4. Extensibility (Spreadability)

The unpaired *t* test for Y max and K parameters comparisons was used. NLCs-F3 were compared with NLCs-F3-L and NLCs-F9 were compared with NLCs-F9-L (Figure 4). Results were expressed as the mean ± SD (standard deviation) in two replicates (*n* = 2). Results were considered statistically significant for *p*-value < 0.05. The Y max suffered a slight increase with linoleic acid but with no statistically significant differences, and no statistically significant differences for K values were found. All the formulations were adjusted for the first order kinetic model.

#### 3.1.5. Rheological Studies

The Newton-mathematical model showed the best statistical fitting of rheological behavior (data not shown). The viscosities measured from the constant share rate period of 50 s^−1^ were 2.197 ± 4.894 × 10^−2^ and 2.306 ± 5.039 × 10^−2^ mPa·s for NLCs-F3 and NLCs-F9, respectively, and 2.222 ± 3.620 × 10^−2^ and 2.236 ± 4.333 × 10^−2^ mPa·s for NLCs-F3-L and NLCs-F9-L, respectively (Figure 5).

### 3.2. Stability Studies

Short-term physical stability was studied from the backscattering profiles obtained by the Turbiscan^®^Lab storing samples at 4, 25, and 37 °C. Unstable formulation was indicated when the backscattering profile presented differences greater than 10% with respect to the initial profile [34]. The BS profiles of selected NLCs revealed variations of less than ±10% at 4 °C and 25 °C, 60 days after production, indicating that the formulations were stable (Appendix A). At 37 °C, there was great variability from 30 days onwards.

Long-term stability of the selected NLCs was also determined according to the International Conference on Harmonization guidelines. The results obtained are shown in Table 2. The EE, Z-Ave, and PI could be maintained without significant changes during a storage period of 60 days at 25 °C and 180 days at 4 °C. However, in the case of storage at 37 °C, the parameters changed starting at day fifteen. ZP decreased over the time in the three storage conditions, and EE was maintained in all cases by over 94%.

### 3.3. In Vitro Release Study

The model that best statistically explained the drug release mechanism in selected NLCs was the First Order. This model presented the smallest value of the AIC (a discriminatory parameter, which is a measure of the best fit based on the maximum likelihood, comparing several models for a given set of data), and it was also confirmed by the value of the coefficient of determination, r^2^ that was closest to one.

First order obtained data were analyzed by the unpaired *t* test to compare NLCs-F3 with respect to NLCs-F3-L; and NLCs-F9 with respect to NLCs-F9-L. *p*-value < 0.05 was set for considering statistically significant differences. No statistically significant differences were found.

As could be observed in Table 3, the release profile of the PF from NLCs-F9 and NLCs-F9-L showed faster release than that from the NLCs-F3 and NLCs-F3-L (Figure 6). However, the maximum concentration of the released drug (Q∞) from NLCs-F3 and NLCs-F3-L was higher in comparison with NLCs-F9 and NLCs-F9-L, but this difference was not statistically significant.

### 3.4. In Vitro Cell Viability Assay

The effect of different concentrations of NLCs-F3-L and NLCs-F9-L solutions on human keratinocytes viability was evaluated using the MTT cytotoxicity assay. After 24 h of incubation, it was observed that dilutions of stock nanoparticles solutions lower than 1/20 significantly reduced cell viability (%) and this cytotoxicity effect increased proportionally with higher concentrations of the NLCs-F3-L and NLCs-F9-L. However, concentrations of PF (μg/mL) below 12 for NLCs-F3-L and 8 for NLCs-F9-L barely affected cell viability (98–100%). A selected NLCs treatment did not affect cell viability, which was close to 100%. Results for the blank treatment were similar to the NLCs incubation. Thus, this result suggested that toxicity was linked to the composition of the blank solution and not to the nanoparticle adding (Figure 7). The presence of surfactants such as Tween 80, necessary for solubilization in an aqueous cellular culture medium, could explain the reduction in cell viability, since it could affect the integrity of cell membranes, and consequently, it increased cell death.

On the other hand, higher concentrations of the blank solution increased the limitation of nutrients, such as glucose or amino acids in the culture medium, which could negatively affect cell viability.

### 3.5. Human Skin Permeation Studies

The permeation profiles of PF with and without penetration enhancers were studied. The cumulative permeated amount of PF (µg) per cm^2^ of human skin in each time interval is represented in Figure 8.

The permeation and prediction parameters of PF with penetration enhancers were calculated. The flow (J) and permeability coefficient (Kp) were determined from the cumulative amount of PF that had permeated through the skin plotted against the time (h) in a steady state.

Table 4 shows that linoleic acid presented the highest values for Qret; thus, it allowed significant retention of PF in the skin after 24 h. All enhancers tested had presented lower values for Css than the therapeutic plasma concentration, so the possible systemic effect can be ruled out or lessened. Therapeutic plasma concentration of PF was 4.89 ± 1.29 µg/mL and 10.19 ± 2.43 µg/mL for young and elderly subjects, respectively.

Selected NLCs formulations permeations were performed on human skin for a period of 24 h, to determine the permeation profile of PF from these formulations, as well as the skin prediction parameters.

NLCs formulations with linoleic acid retained a higher amount of PF in skin than formulations without linoleic acid, (Table 5).

Table 6 exhibits the results obtained from the predicted steady-state plasma concentration of PF after the application of selected NLCs on 100 cm^2^ of skin. All the Css values obtained for the selected formulations were below the therapeutic plasma concentration.

### 3.6. Histological Analysis

To examine the anti-inflammatory effect of the NLCs-F3-L and NLCs-F9-L formulations, the mice were subjected to the topical application of Arachidonic acid (AA) on the ear to model skin inflammation. AA induced ear inflammation in positive control mice producing redness and edema. Histopathological images after 1 h of treatment are shown in Figure 9. Control ear micrographs consisted of a relatively thin epidermis with a contiguous stratum corneum and dermis. AA application led to a significant increase in the epidermis and dermis thickness with the loss of stratum corneum, which was accompanied by the presence of inflammatory cells and a loosening of the connective tissue. Loss of the stratum corneum was also evident in the PF treated mice, along with a leucocyte infiltrate (arrowhead, Figure 9C). When the ears were treated with NLCs-F3-L and NLCs-F9-L, a better profile was observed with less inflammatory cells infiltrates. Moreover, the mice treated with NLCs-F9-L exhibited a lower epidermis thickness and the stratum corneum was similar to that of the control conditions, whereas in the case of NLCs-F3-L, a little hyperplasia of the epidermis and some loss of the stratum corneum were observed.

### 3.7. Model of Mice Ear Inflammation Induced with TPA

Anti-inflammatory activity of the selected NLCs was evaluated, using TPA (12-O-tetradecanoylphorbol 13-acetate) to induce mouse ear inflammation. The selected formulations showed good results of the anti-inflammatory efficacy studies and the formulations showed a significant reduction of the dermal edema compared to the indomethacin. As shown in Figure 10, the inhibition of inflammation was significantly higher in the formulations with linoleic acid (NLCs-F3-L and NLCs-F9-L). NLCs-F3-L was the formulation with the best anti-inflammatory efficacy.

### 3.8. In Vivo Mice Model and Inflammatory Response after the NLCs-F3-L and NLCs-F9-L Treatment

The inflammatory process associated with the NLCs-F3-L and NLCs-F9-L treatment in an in vivo ear mice model of inflammation, induced by arachidonic acid, was evaluated by the RT-qPCR of different proinflammatory cytokines genes, such as IL-6, IL-1β, and TNF-α, and IL-10 as the anti-inflammatory cytokine [35]. Mice ears were treated with 50 µL of arachidonic acid 0.5% for 15 min to induce the inflammation, and then the ears were treated with 0.5 mL of NLCs-F3-L or NLCs-F9-L, containing 356 and 235 µg of PF, respectively, or 0.5 mL of blank solution.

Gene expression results shown in Figure 11 confirmed the inflammatory effect of arachidonic acid (control +) compared with the non-treated ears (control −), with a significative increase in the expression of the proinflammatory genes IL-6, IL-1β, and TNF-α. However, after the NLCs-F3-L and NLCs-F9-L treatment, these inflammatory levels were significantly reduced for all the proinflammatory genes studied. In parallel, swollen ears were also treated with a blank solution that did not contain nanoparticles, which did not affect the proinflammatory cytokine results, compared with the control +, confirming the anti-inflammatory effect of the NLCs-F3-L and NLCs-F9-L.

Regarding the anti-inflammatory cytokine IL-10, the control + showed a slight tendency reduction of expression without significative differences compared with the non-treated mice (control −). In this case, the treatment with NLCs-F3-L and NLCs-F9-L induced a small non-significant increase in IL-10 expression compared with the control +. These results were because anti-inflammatory IL-10 required longer exposure to arachidonic acid and NLCs incubation to produce significative changes in the mRNA level expression [36].

In view of these results, it could be considered a positive potential effect of NLCs structures that they can act as inner vehicles to transport PF to the inflammation sites in a safe and efficient way.

### 3.9. Skin Integrity Parameters

The evolution of biomechanical parameters (TEWL and SCH) before and after the application of formulations assayed is shown in Figure 12. Slight but statistically significant decreases in the TEWL values for selected NLCs were recorded, which explained an occlusive effect without altering skin integrity. However, the SCH suffered a slight but not statistically significant increase. Given that skin capacitance is directly related to skin hydration, these results indicated that the formulations did not increase the hydration with respect the normal behavior of the skin.

### 3.10. Skin Irritation Testing

Appendix A shows the in vivo Draize skin test of the selected NLCs that was assessed in male albino rabbits. No erythema nor edema were found after 24, 48, and 72 h of exposure, and no signs of skin irritancy were detected, since the individual primary irritancy index determined for these formulations was less than 0.5.

## 4. Discussion

The use of nanocarriers to localize the drug in the inflamed areas of the skin can be considered of great importance in limiting the side effects of drugs on the uninvolved healthy skin areas, as well as the systemic circulation.

Nanostructured materials are very promising for the treatment of inflammatory skin conditions. They can modulate the delivery of the active ingredients to different layers of the skin and can selectively target the diseased areas or promoting cells [5].

The selected formulations were chosen from the factorial design depicting the best properties in an attempt to optimize the formulation [37,38]. The ZP is a measure of the particle charge and can influence both the stability of the particle and its mucoadhesion. Electrostatic repulsion between particles with the same polarity of electrical charge prevents aggregation [11]. All the formulations had a net negative charge.

Figure 1 shows TEM images and confirms the adequacy of the preparation method. It is well known that the smaller the particles the higher the particles adhesiveness to the surface such as tissues [32].

To develop such controlled-release formulations for application to the skin, rheology or viscosity is a critical parameter to consider. The constant viscosity when the shear rate increased was indicative of Newtonian behavior by exhibiting low values of viscosity [39,40].

Backscattering profiles were used for the evaluation of short-term physical stability of the formulations. The BS profiles of the selected NLCs revealed variations of less than ±10% at 4 and 25 °C, 60 days after production, indicating that the formulations were stable. The slight destabilization observed in the formulation profiles was reversible. This stability was also confirmed by long-term stability under storage at 4 and 25 °C. The EE, Z-Ave, and PI could be maintained without significant changes over a storage period of 60 days at 25 °C and 180 days at 4 °C. However, in the case of storage at 37 °C, changes were evident within 15 days.The release profile of the drug from the vehicle is important information that can be used to predict in vivo behavior [41]. As expected for nanostructured lipid carriers, a slow and sustained release profile for optimized NLCs was observed, explained by the solid matrix of this nanostructure and the subsequent drug immobilization. This was an indicator of the excellent suitability of this vehicle for PF.

The appropriateness of the selected NLCs kinetic model fitting was determined by the calculation of the r^2^ and AIC [42], which was of the best fit based on the maximum likelihood. First order was the model associated with the smallest value of the AIC and it was regarded as providing the best fit out of the set of available models. All the Css values obtained for the assayed formulations were below the therapeutic plasma concentration. These results might ensure that the topical application of these formulations would not have any systemic effect and ensure a local anti-inflammatory and analgesic effect of the drug as described by Abrego et al. [42].The HaCaT cell line is a common *in vitro* skin model that has been widely used to test the cell cytotoxicity of different nanoparticles [43,44]. In this study, the cells were incubated for 24 h with different concentrations of optimized NLCs and Blank-NLCs as a reference to compare. Results from Figure 7 show that Blank-NLCs (not loaded with PF) had the same cytotoxicity profile than those carrying the anti-inflammatory compound, which indicates that the cellular damage was not due to the PF concentration if not the NLCs composition. This result agreed with previous studies performed by Graham et al. [45], where the authors assayed higher concentrations of nanoparticles carrying similar anti-inflammatory components using the same HaCaT model.

On the other hand, when the different dilutions of NLCs-F3-L, NLCs-F9-L, and Blank-NLCs were tested, it was observed, for all nanoparticles types, that samples diluted at more than 1/50 barely affected cell viability, with a 95% cell survival. However, when the HaCaT were incubated with nanoparticles samples concentrated more than 1/20, it negatively affected cell viability in a concentration-dependent manner. This result confirmed that maybe one or some components of the NLCs at high concentrations were responsible for the cell death. Thus, only the PF concentrations from 0.3 to 12 µg/mL for NLCs-F3-L, and from 0.2 to 8 µg/mL for NLCs-F9-L, should be used as suitable for topical use, due to the absence of toxicity.

Anti-inflammatory activity of the selected NLCs was evaluated using a topical inflammatory model. The values obtained for the selected NLCs were higher, compared with the results obtained previously for PF polymeric nanoparticles described by Abrego [8]. The NLCs process of permeation through the skin takes longer than the polymeric nanoparticles and it leads to the increased contact of the drug with the dermis. The lipid nanoparticles dispersion takes the transcellular route to penetrate through the skin and the outstanding advantage is the easy and complete biodegradation of the lipid nanoparticles. Lipids as glycerides are natural materials and are easily degraded by natural processes such as enzymes. However, this is a crucial and often discussed aspect of the cytotoxicity of the polymers after internalization into cells [46].

The cutaneous permeability barrier is mediated by extracellular lipids, mainly ceramides, free fatty acids, and cholesterol, which form extracellular lipid-enriched lamellar membranes between the corneocytes that block the movement of water and the electrolytes [2]. NLCs have gained attention as particulate systems to improve the delivery of lipophilic drugs due to the high affinity of these molecules for the lipid matrix [47]. As a general result, it could be considered that the selected NLCs did not alter the biophysical properties of the skin, and therefore, were suitable and safe for topical application. TEWL is an important indicator of skin integrity. Any increase in the value of the TEWL is closely related to alterations in the skin barrier function [48]. All participants showed TEWL values that were situated in the normal range. Slight but statistically significant decreases in the TEWL values for the selected NLCs was recorded, which explained an occlusive effect without altering skin integrity. The NLCs form an adhesive layer occluding the skin surface, as is typical for lipid particles [49]. In all cases, skin hydration remained unaffected by the formulation components. These results agreed with the results obtained for the histological studies.

The irritancy test was performed in male albino rabbits. The possibility of causing skin damage is of vital importance in the development of topical treatments [32]. No signs of skin irritancy were detected, since the individual primary irritancy indexes determined for these formulations were less than 0.5, this was good evidence of skin tolerability, assigned predominantly to the use of biodegradable, well tolerated, and physiological excipients especially developed for application on the skin [50].

The present study provides evidences that dermal application of the selected NLCs formulations could be an effective system for the delivery and controlled release of PF, improving the biopharmaceutical profile of this drug, facilitating the contact of the PF on the skin and improving its dermal retention, and it also reduced the dermal oedema of this drug. It could be of interest for the development of future treatment of inflammatory skin disease.

Significant thickening of the skin and marked accumulation of inflammatory cells were observed in the AA treated mice ear. Treatment with NLCs-F3-L and NLCs-F9-L resulted in a remarkable reduction of the number of neutrophils infiltrated. Surprisingly, almost no inflammatory cells were observed after the treatment with NLCs-F9-L. However, in the case of NLCs-F3-L, there was a little hyperplasia of the epidermis and the loss of stratum corneum. In contrast, mice treated with conventional PF still showed marked accumulation of inflammatory cells. These results demonstrated that the novel NLCs-F9-L showed a better therapeutic potential for the inflammatory process as compared to the current PF formulation.

## Figures and Tables

**Figure 1 nanomaterials-08-01022-f001:**
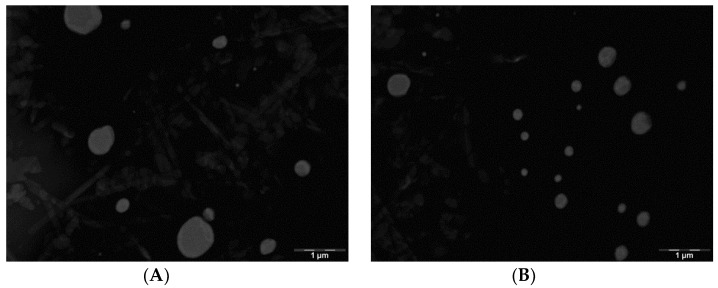
Transmission Electron Microscopy image of NLCs F3-L (**A**) and NLC F9-L (**B**).

**Figure 2 nanomaterials-08-01022-f002:**
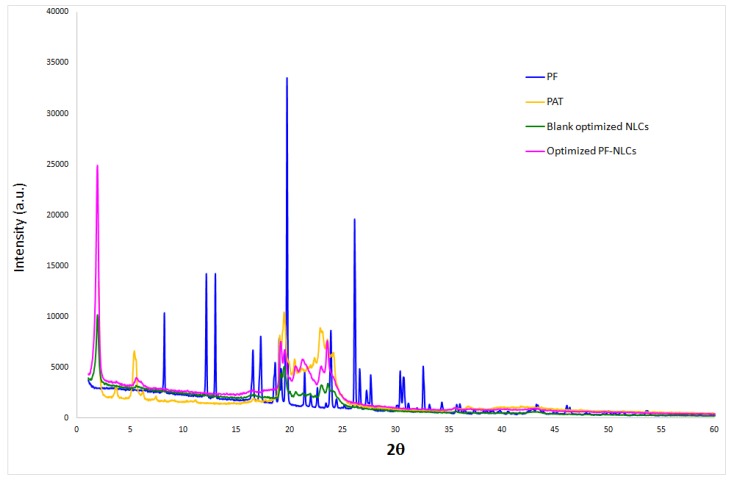
X-Ray diffraction patterns.

**Figure 3 nanomaterials-08-01022-f003:**
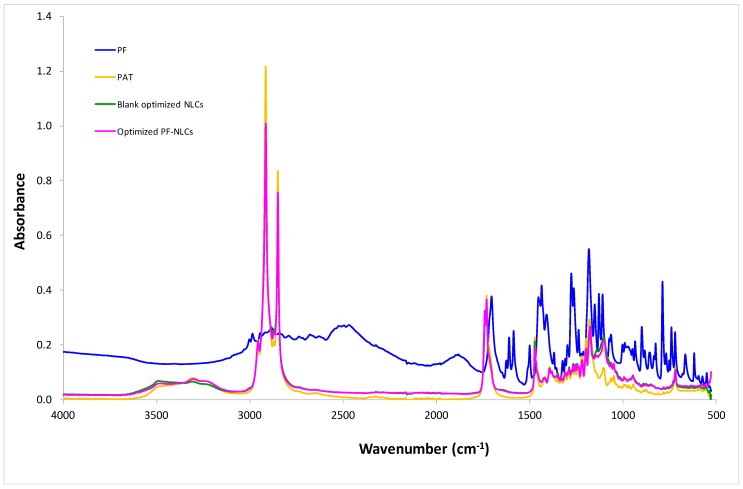
Fourier Transform Infrared (FTIR) analysis.

**Figure 4 nanomaterials-08-01022-f004:**
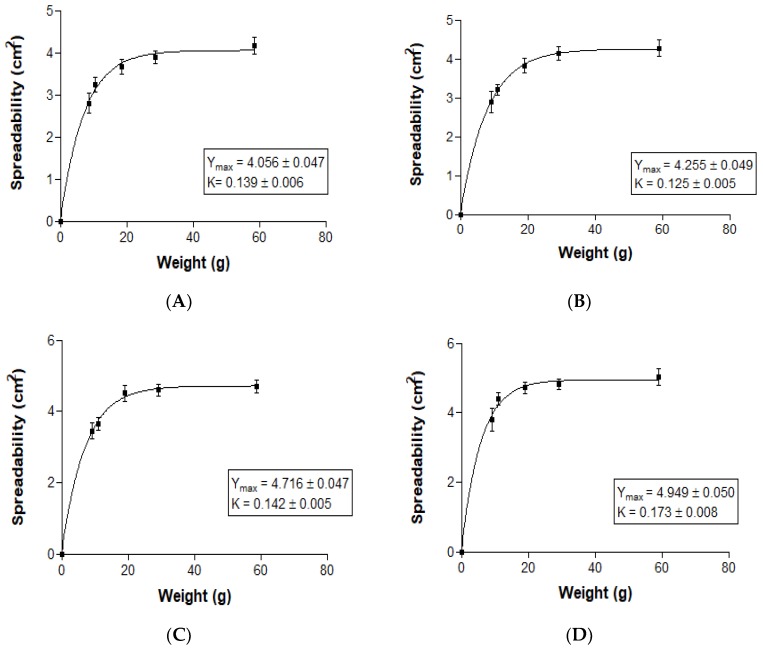
Spreadability of NLCs-F3 (**A**); NLCs-F3-L (**B**); NLCs-F9 (**C**); and NLCs-F9-L (**D**).

**Figure 5 nanomaterials-08-01022-f005:**
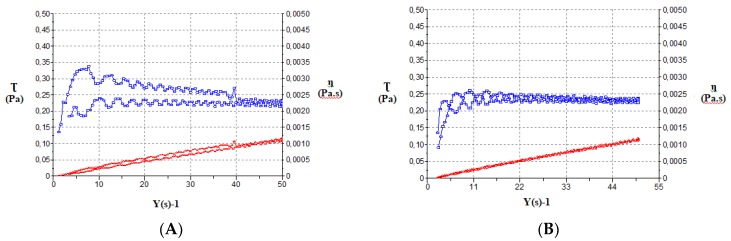
Rheological behavior of NLCs-F3 (**A**); NLCs-F9 (**B**); NLCs-F3-L (**C**); NLCs-F9-L (**D**).

**Figure 6 nanomaterials-08-01022-f006:**
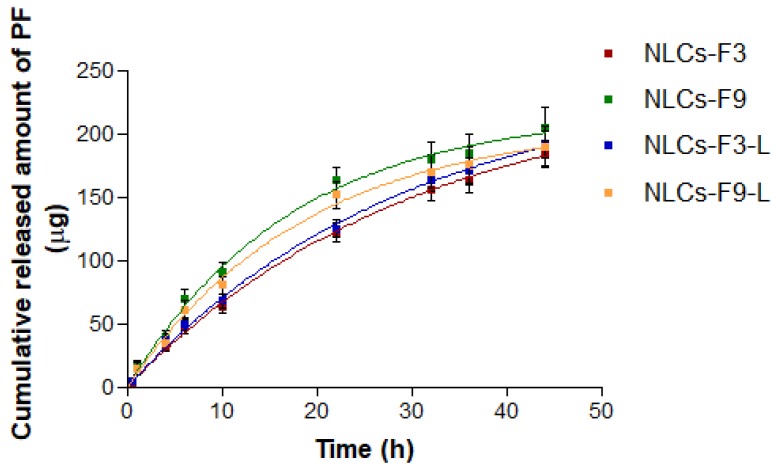
In vitro Pranoprofen (PF) release profiles from the optimized nanostructured lipid carriers. The cumulative amount was plotted versus time. Data are represented as mean ± SD (*n* = 3).

**Figure 7 nanomaterials-08-01022-f007:**
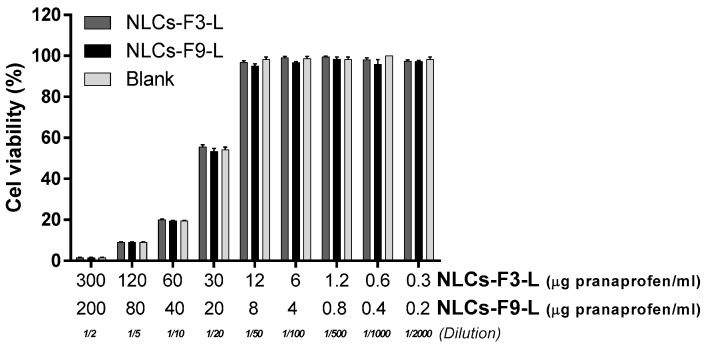
Cell viability percentage of HaCaT cells treated with different concentrations of NLCs-F3-L and NLCs-F9-L after 24 h of incubation. Results are shown as mean ± SEM of 3 independent experiments (*p* < 0.05).

**Figure 8 nanomaterials-08-01022-f008:**
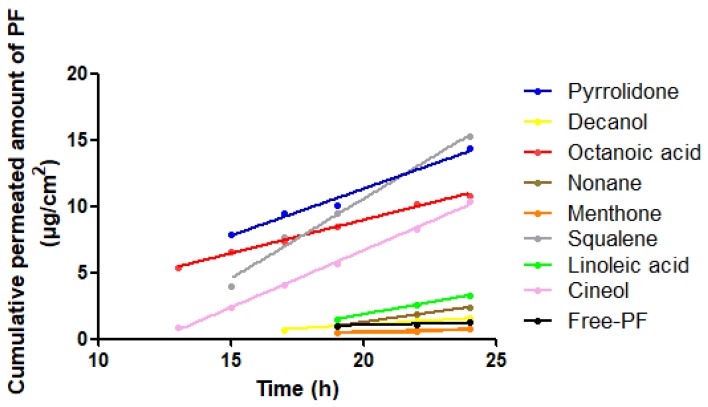
Median cumulative permeated amount of PF with and without penetration enhancers versus time (h) through human skin.

**Figure 9 nanomaterials-08-01022-f009:**
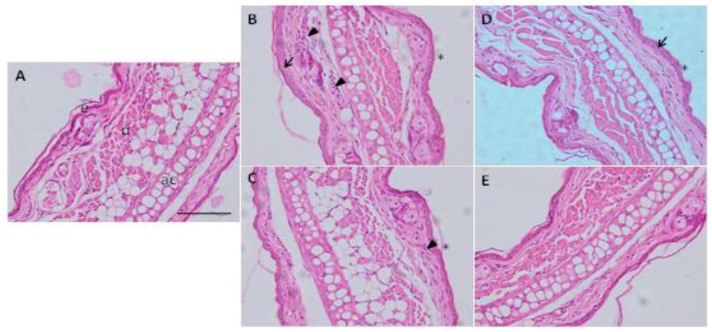
Representative micrographs of mice ear (x200 magnification). (**A**) = Non-inflamed ear, (**B**) = Positive control (inflamed ear with arachidonic acid), (**C**) = pranoprofen alone, (**D**) = pranoprofen (0.75 mg/mL) + linoleic acid (5%) (NLCs-F3-L), (**E**) = pranoprofen (0.5 mg/mL) + linoleic acid (5%) (NLCs-F9-L). Scale bar = 100 µM. e = epidermis, d = dermis, ac = auricular cartilage. Arrowheads indicate neutrophilic infiltrates, arrows indicate increased epidermis, and asterisks indicate loss of stratum corneum.

**Figure 10 nanomaterials-08-01022-f010:**
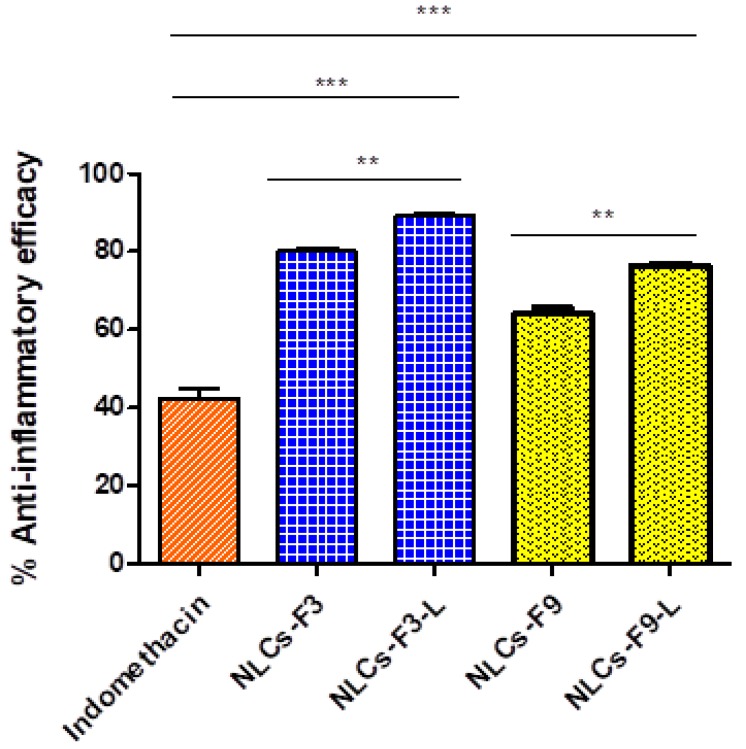
In vivo anti-inflammatory efficacy after TPA (12-*O*-tetradecanoylphorbol 13-acetate) induced mouse edema. Mean ± SD (n = 6). Significant statistical differences: ** *p* < 0.01, *** *p* < 0.001.

**Figure 11 nanomaterials-08-01022-f011:**
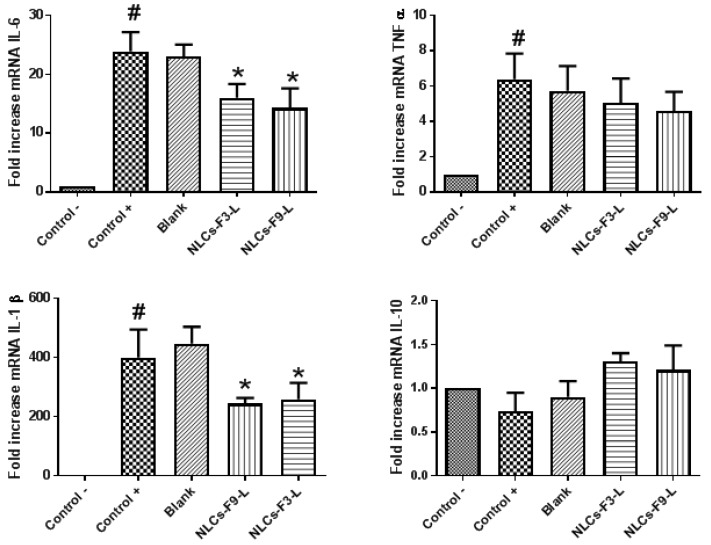
Relative mRNA level expression of inflammatory biomarkers IL-6, TNF-α, IL-1β, and IL-10 in mice ear tissue after treatment with NLCs-F3-L and NLCs-F9-L. Differences between control − (non-treated skin) and control + (arachidonic treatment) is shown as #. Differences between control + and blank, NLCs-F3-L or NLCs-F3-L is shown as *. Results are shown as mean ± SEM of 3 independent experiments (*p* < 0.05).

**Figure 12 nanomaterials-08-01022-f012:**
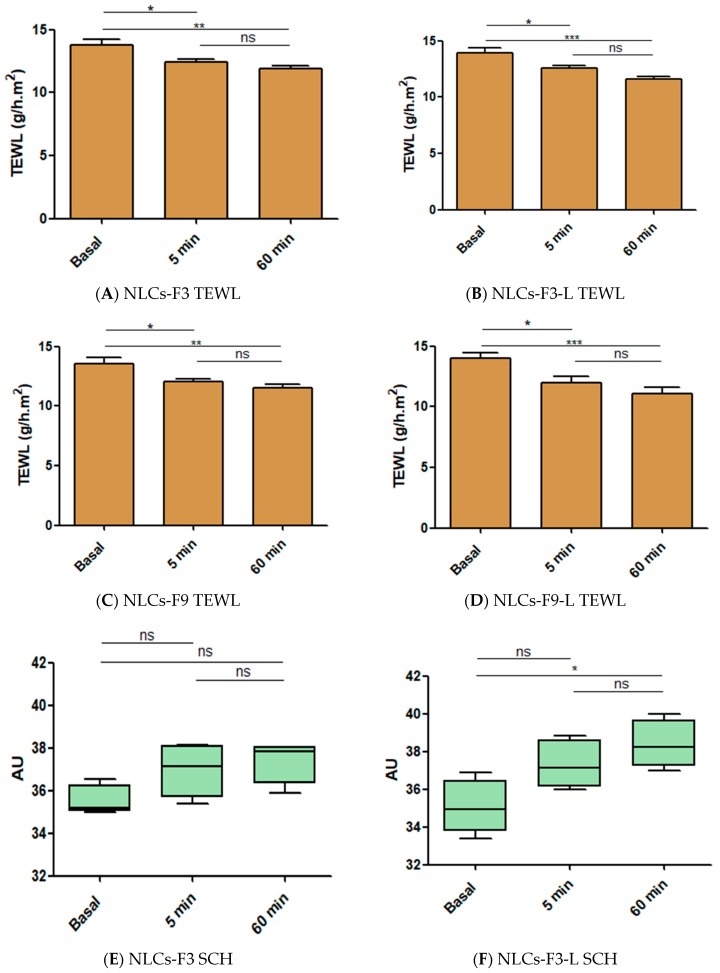
Biomechanical parameters evolution monitored before the application of the formulations and 1 h after application. TEWL is expressed as g/h × cm^2^ and the SCH as arbitrary units (AU). Significant statistical differences: * *p* < 0.05, ** *p* < 0.01, *** *p* < 0.001, ns = non-significant.

**Table 1 nanomaterials-08-01022-t001:** Composition and physicochemical characterization of selected nanostructured lipid carriers (NLCs).

	Composition				Physicochemical Characterization			
	cPF (%)	cSL/L (%)	cTW (%)	Linoleic Acid (%)	Z-Ave (nm) ± SD	PI ± SD	ZP (mV) ± SD	EE (%) ± SD
NLCs-F3	1.50	50.00	2.50	-	248.40 ± 6.34	0.229 ± 0.046	−10.70 ± 0.44	99.68 ± 0.17
NLCs-F3-L	1.50	50.00	2.50	5	274.59 ± 8.37	0.255 ± 0.077	−9.89 ± 0.50	97.96 ± 0.24
NLCs-F9	1.00	76.80	3.00	-	222.30 ± 9.74	0.237 ± 0.085	−9.20 ± 0.48	99.61 ± 0.12
NLCs-F9-L	1.00	76.80	3.00	5	266.98 ± 9.82	0.280 ± 0.039	−10.01 ± 0.36	98.37 ± 0.28

**Table 2 nanomaterials-08-01022-t002:** Stability of the selected NLCs at different storage conditions.

**A NLCs-F3**					
**Time (days)**	**Temperature**	**Z-Ave (nm) ± SD**	**PI± SD**	**ZP (mV) ± SD**	**EE (%)**
**1**	4 °C	249.00 ± 4.95	0.29 ± 0.05	−10.63 ± 0.08	99.62
**3**		249.60 ± 8.85	0.30 ± 0.04	−10.75 ± 0.48	99.59
**8**		265.77 ± 8.77	0.29 ± 0.02	−10.46 ± 0.17	99.08
**15**		269.30 ± 3.23	0.24 ± 0.02	−9.84 ± 0.16	98.90
**30**		278.80 ± 5.87	0.33 ± 0.04	−8.99 ± 0.64	98.86
**60**		290.47 ± 3.33	0.35 ± 0.06	−9.05 ± 0.54	98.79
**90**		306.52 ± 6.02	0.27 ± 0.08	−9.00 ± 0.26	98.23
**180**		314.99 ± 5.77	0.30 ± 0.08	−8.98 ± 0.34	98.00
**1**	25 °C	248.65 ± 5.12	0.23 ± 0.04	−10.62 ± 0.29	99.68
**3**		249.60 ± 6.40	0.23 ± 0.03	−10.53 ± 0.34	99.07
**8**		279.87 ± 5.59	0.25 ± 0.02	−10.60 ± 0.33	98.33
**15**		308.43 ± 4.15	0.23 ± 0.05	−10.68 ± 0.40	98.68
**30**		325.20 ± 6.86	0.26 ± 0.02	−10.16 ± 0.12	98.84
**60**		378.30 ± 3.68	0.20 ± 0.03	−9.80 ± 0.15	98.06
**90**		410.38 ± 4.01	0.33 ± 0.06	−9.33 ± 0.26	98.02
**180**		478.40 ± 6.25	0.28 ± 0.03	−9.35 ± 0.19	97.38
**1**	37 °C	253.77 ± 4.66	0.27 ± 0.03	−10.59 ± 0.41	99.67
**3**		499.54 ± 6.34	0.28 ± 0.02	−10.60 ± 0.22	99.66
**8**		700.45 ± 6.67	0.27 ± 0.05	−10.30 ± 0.28	98.01
**15**		1134.22 ± 29.62	0.39 ± 0.09	−9.99 ± 0.36	98.01
**30**		1998.05 ± 26.89	0.45 ± 0.08	−10.50 ± 0.35	97.60
**60**		2467.20 ± 12.11	0.45 ± 0.05	−10.42 ± 0.63	97.43
**90**		3001.06 ± 49.94	0.47 ± 0.03	−10.51 ± 0.50	97.33
**180**		-	-	−10.56 ± 0.21	96.98
**B NLCs-F3-L**					
**Time (days)**	**Temperature**	**Z-Ave (nm) ± SD**	**PI ± SD**	**ZP (mV) ± SD**	**EE (%)**
**1**	4 °C	225.30± 8.02	0.29 ± 0.03	−9.34 ± 0.11	99.03
**3**		256.44± 6.95	0.29 ± 0.04	−9.35 ± 0.48	98.98
**8**		268.90 ± 9.75	0.31 ± 0.03	−9.16 ± 0.07	99.03
**15**		273.13 ± 5.10	0.33 ± 0.06	−8.48 ± 0.20	98.85
**30**		276.60 ± 7.92	0.34 ± 0.05	−8.81 ± 0.76	98.84
**60**		280.30 ± 3.15	0.29 ± 0.06	−8.21 ± 0.53	98.86
**90**		296.72 ± 6.02	0.28 ± 0.07	−8.40 ± 0.36	98.12
**180**		299.98 ± 4.68	0.33 ± 0.06	−8.33 ± 0.31	97.35
**1**	25 °C	222.90 ± 4.94	0.24 ± 0.01	−9.80 ± 0.28	98.93
**3**		256.54 ± 4.70	0.25 ± 0.03	−9.39 ± 0.25	98.84
**8**		280.82 ± 2.40	0.24 ± 0.02	−9.86 ± 0.23	98.80
**15**		307.00 ± 4.15	0.24 ± 0.03	−9.68 ± 0.34	98.87
**30**		328.10 ± 6.86	0.28 ± 0.04	−9.16 ± 0.12	98.73
**60**		357.20 ± 2.89	0.29 ± 0.02	−9.12 ± 0.15	98.91
**90**		400.04 ± 5.01	0.27 ± 0.07	−8.19 ± 0.22	98.80
**180**		423.56 ± 8.01	0.28 ± 0.05	−8.51 ± 0.16	97.08
**1**	37 °C	223.50 ± 3.97	0.29 ± 0.03	−9.90 ± 0.43	98.99
**3**		497.70 ± 6.76	0.31 ± 0.02	−9.10 ± 0.23	99.07
**8**		703.11 ± 7.17	0.29 ± 0.05	−10.30 ± 0.36	98.95
**15**		1019.00 ± 22.32	0.30 ± 0.03	−9.45 ± 0.32	98.87
**30**		1654.86 ± 15.21	0.38 ± 0.04	−9.48 ± 0.14	99.02
**60**		2007.67 ± 9.94	0.37 ± 0.06	−10.00 ± 0.14	98.57
**90**		2700.01 ± 44.12	0.39 ± 0.07	−10.10 ± 0.25	98.95
**180**		5002.07 ± 234.45	0.48 ± 0.06	−9.30 ± 0.43	97.02
**C NLCs-F-9**					
**Time (days)**	**Temperature**	**Z-Ave (nm) ± SD**	**PI ± SD**	**ZP (mV) ± SD**	**EE (%)**
**1**	4 °C	242.07 ± 3.93	0.24 ± 0.03	−10.48 ± 0.27	99.06
**3**		242.89 ± 6.30	0.26 ± 0.02	−10.85 ± 0.50	98.82
**8**		260.90 ± 9.75	0.30 ± 0.05	−9.86 ± 0.16	99.12
**15**		276.10 ± 2.72	0.26 ± 0.03	−9.48 ± 0.09	98.81
**30**		257.02 ± 5.42	0.27 ± 0.02	−8.71 ± 0.22	98.80
**60**		263.30 ± 1.25	0.29 ± 0.02	−9.21 ± 0.19	98.78
**90**		300.74 ± 5.69	0.29 ± 0.06	−9.00 ± 0.23	98.50
**180**		320.22 ± 7.43	0.33 ± 0.05	−8.12 ± 0.29	97.35
**1**	25 °C	240.75 ± 3.69	0.25 ± 0.03	−10.70 ± 0.23	98.99
**3**		249.55 ± 6.21	0.25 ± 0.01	−10.61 ± 0.24	98.82
**8**		286.52 ± 4.02	0.27 ± 0.02	−10.62 ± 0.30	98.79
**15**		299.06 ± 3.84	0.26 ± 0.02	−10.54 ± 0.34	98.88
**30**		310.09 ± 5.18	0.27 ± 0.01	−9.76 ± 0.22	98.84
**60**		339.56 ± 5.16	0.27 ± 0.01	−9.68 ± 0.20	98.81
**90**		411.47 ± 4.02	0.28 ± 0.08	−9.50 ± 0.18	98.81
**180**		459.60 ± 6.20	0.29 ± 0.05	−9.32 ± 0.14	97.08
**1**	37 °C	244.99 ± 2.55	0.24 ± 0.02	−10.60 ± 0.47	99.00
**3**		487.54 ± 8.04	0.25 ± 0.01	−10.68 ± 0.22	99.06
**8**		698.99 ± 8.89	0.29 ± 0.06	−10.55 ± 0.33	98.84
**15**		1076.04 ± 5.52	0.30 ± 0.02	−9.97 ± 0.36	98.89
**30**		1673.13 ± 17.08	0.33 ± 0.04	−10.18 ± 0.21	99.02
**60**		2197.46 ± 9.89	0.34 ± 0.02	−10.11 ± 0.26	98.63
**90**		2998.34 ± 44.19	0.39 ± 0.05	−10.30 ± 0.32	98.74
**180**		-	-	−11.11 ± 0.57	96.08
**D NLCs-F9-L**					
**Time (days)**	**Temperature**	**Z-Ave (nm) ± SD**	**PI ± SD**	**ZP (mV) ± SD**	**EE (%)**
**1**	4 °C	232.62 ± 3.20	0.28 ± 0.02	−9.74 ± 0.23	98.97
**3**		239.90 ± 1.97	0.30 ± 0.03	−9.85 ± 0.44	98.61
**8**		242.76 ± 5.23	0.27 ± 0.04	−9.06 ± 0.19	98.94
**15**		246.10 ± 2.72	0.28 ± 0.01	−8.98 ± 0.20	98.80
**30**		253.80 ± 5.49	0.31 ± 0.02	−8.71 ± 0.22	98.89
**60**		259.57 ± 3.09	0.29 ± 0.05	−8.61 ± 0.29	98.60
**90**		280.83 ± 5.11	0.32 ± 0.03	−8.20 ± 0.25	98.67
**180**		301.85 ± 3.86	0.35 ± 0.07	−7.98 ± 0.37	98.10
**1**	25 °C	231.54 ± 2.22	0.25 ± 0.02	−10.00 ± 0.37	98.95
**3**		246.81 ± 3.18	0.24 ± 0.03	−9.98 ± 0.25	98.75
**8**		260.51 ± 2.66	0.26 ± 0.02	−9.96 ± 0.22	98.75
**15**		307.00 ± 4.15	0.27 ± 0.01	−9.88 ± 0.29	99.08
**30**		328.10 ± 6.86	0.29 ± 0.02	−9.05 ± 0.13	98.89
**60**		333.20 ± 3.68	0.30 ± 0.01	−9.12 ± 0.16	98.78
**90**		410.64 ± 2.97	0.29 ± 0.06	−9.00 ± 0.18	98.50
**180**		497.70 ± 5.75	0.32 ± 0.03	−8.80 ± 0.16	97.39
**1**	37 °C	243.60 ± 4.96	0.27 ± 0.02	−9.80 ± 0.39	98.99
**3**		300.70 ±.3.01	0.30 ± 0.02	−9.80 ± 0.23	99.02
**8**		676.23 ± 5.88	0.27 ± 0.02	−9.92 ± 0.36	98.89
**15**		1186.12 ± 21.03	0.28 ± 0.02	−9.70 ± 0.26	98.88
**30**		1967.03 ± 22.72	0.29 ± 0.02	−9.86 ± 0.15	99.00
**60**		2401.35 ± 23.00	0.30 ± 0.04	−10.01 ± 0.34	98.58
**90**		2884.10 ± 74.09	0.39 ± 0.03	−9.76 ± 0.25	97.94
**180**		-	0.41 ± 0.04	−9.80 ± 0.52	94.32

**Table 3 nanomaterials-08-01022-t003:** *In vitro* release profiles. Mean parameters obtained after fitting the release data from NLCs-F3, NLCsF3-L, NLCs-F9, and NLCs-F9-L (mean ± SD, *n* = 3) to different Kinetic model equations.

Order Equation		Parameters	Unit	Value (Mean ± SD)			
				NLCs-F3	NLCs-F3-L	NLCs-F9	NLCs-F9-L
Zero order	Q_t_ = K_0_ t + Q_∞_	K_0_	µg/h	4.09 ± 0.34	4.21 ± 0.33	4.30 ± 0.56	4.10 ± 0.54
		Q_∞_	µg	17.48 ± 8.21	19.76 ± 8.26	35.11 ± 13.94	30.19 ± 13.26
		r^2^	-	0.9539	0.9559	0.8881	0.8887
		AIC	-	71.53	71.61	79.92	79.11
First order	Q_t_ = Q_∞_(1-e^−kf·t^)	K_f_	h^−1^	0.032 ± 0.009	0.034 ± 0.009	0.057 ± 0.015	0.053 ± 0.015
		Q_∞_	µg	239.20 ± 38.62	244.00 ± 36.70	217.7 ± 22.89	209.30 ± 25.63
		r^2^	-	0.9752	0.9757	0.9472	0.9438
		t ½	h	21.16	20.38	12.15	13.07
		AIC	-	66.55	66.87	73.92	73.65
Korsmeyer-Peppas	Q_t_ = K_k_ t ^n^	K_K_	h^−n^	13.17± 3.27	14.42 ± 3.31	23.63 ± 6.96	20.47 ± 6.56
		n	-	0.70 ± 0.07	0.69 ± 0.07	0.58 ± 0.09	0.60 ± 0.09
		r^2^	-	0.9727	0.9752	0.9323	0.9284
		AIC	-	67.32	67.02	75.89	75.59
Weibull	Q_t_ = Q_∞_(1-e^−(t/td)β^)	t_d_	h	39.80 ± 25.51	52.50 ± 47.64	17.28 ± 4.94	17.55 ± 4.86
		β	-	0.924 ± 0.238	0.859 ± 0.232	1.003 ± 0.287	1.048 ± 0.307
		Q_∞_	µg	276.2 ± 97.5	333.3 ± 165.7	217.2 ± 26.7	202.7 ± 24.9
		r^2^	-	0.9756	0.9769	0,9472	0.9440
		AIC	-	68.43	68.43	75.92	75.62

**Table 4 nanomaterials-08-01022-t004:** Skin permeation parameters of PF in the presence of tested penetration enhancers. Data are represented as median (min-max).

Penetration Enhancers	J (µg/(h/cm^2^))	K_p_ (cm/h) × 10^5^	Q_ret_ (µg/g/cm^2^)	Q_24h_ (µg)	Young Subject Css (µg/mL) × 10^3^	Elderly Subject Css (µg/mL) × 10^3^
Pyrrolidone	0.780 ^i^ (0.600–0.800)	111.0 ^i^ (94.0–125.0)	94.61 ^i^ (89.00–99.00)	13.00 ^i^ (12.10–14.32)	96.41 (81.75–109.00)	181.54 (153.94–205.25)
Decanol	0.121 ^a,c,d,e,f,g,h,i^ (0.117–0.120)	18.8 ^a,c,d,e,f,g,h,i^ (18.3–18.9)	107.41 ^a,c,d,e,f,g,h,i^ (103.00–113.01)	11.18 ^a,c,d,e,f,g,i^ (10.00–12.00)	16.35 (15.99–16.47)	30.78 (30.12–31.02)
Octanoic acid	0.534 ^a,d,e,f,g,h,i^ (0.400–0.600)	83.5 ^a,d,e,f,g,h^ (62.5–93.8)	130.78 ^a,d,e,f,g,h,i^ (124.01–135)	87.64 ^a,d,e,f,g,h,I^ (86.30–88.80)	72.82 (54.50–81.75)	137.13 (102.65–153.93)
Nonane	0.841 ^e,f,g,i^ (0.700–0.950)	130.0 ^e,f,g,i^ (110.0–150.0)	65.22 ^a,f,g,h,i^ (62.00–68.02)	13.02 ^e,f,g,h,i^ (12.40–14.50)	114.59 (95.37–129.44)	215.77 (179.59–243.73)
Menthone	0.051 ^a,f,g,h^ (0.400–0.600)	7.8 ^a,f,g,h,i^ (6.3–9.4)	69.21 ^a,f,g,h,i^ (64.00–73.00)	3.91 ^a,g,h,i^ (2.70–4.60)	6.84 (5.45–8.17)	12.88 (10.26–15.39)
Squalene	1.203 ^a,g,h,i^ (1.100–1.300)	190.0 ^a,g,h,i^ (170.0–200.0)	40.81 ^a,g,h,i^ (37.00–45.02)	15.22 ^a,g,h,i^ (13.90–16.30)	163.91 (149.87–177.12)	308.64 (282.22–333.53)
Linoleic acid	0.362 ^a,i^ (0.200–0.400)	56.6 ^a,i^ (31.3–62.5)	194.99 ^a,h,i^ (190.00–197.00)	3.31 ^a,h,i^ (2.10–4.60)	49.34 (27.25–54.50)	92.90 (51.31–102.62)
Cineol	0.860 ^i^ (0.700–0.900)	130.0 ^i^ (110.0–140.0)	92.81 ^a,i^ (86.00–97.00)	10.35 ^a,i^ (9.10–11.20)	117.17 (95.37–122.62)	220.64 (179.59–230.90)
Free-PF	0.050 (0.040–0.060)	79.0 (63.0–94.0)	35.04 (31.30–36.21)	1.23 (0.98–1.33)	68.63 (54.50–81.75)	129.23 (102.62–153.94)

^a^ Pyrrolidone; ^b^ Decanol; ^c^ Octanoic acid; ^d^ Nonane; ^e^ Menthone; ^f^ Squalene; ^g^ Linoleic acid; ^h^ Cineol; ^i^ Free-PF. Data are represented as median (min-max). Mann-Whitney U Tests were performed to assess the statistical significance differences (*p* < 0.05) for Jss, Kp, Qret and Q24h parameters.

**Table 5 nanomaterials-08-01022-t005:** Statistical differences between NLCs-F3 and NLCs-F3-L, and between NLCs-F9 and NLCs-F9-L. Results are reported as the median value and minimum-maximum range values (*n* = 6). Letters represent significant differences (*p* < 0.05) between ^a^ NLCs-F3 and NLCs-F3-L, and between ^b^NLCs-F9 and NLCs-F9-L.

Parameters	NLCs-F3	NLCs-F3-L	NLCs-F9	NLCs-F9-L
J × 10^2^ mg/cm^2^h	7.27 ^a^ (6.30–8.51)	13.17 (11.14–14.27)	7.45 ^b^ (6.38–8.48)	9.11 (7.99–10.47)
Kp × 10^4^ (cm/h)TL (h) Q_R_ (µg/cm^2^g)	1.6 ^a^ (1.4–1.9) 1.67 ^a^ (1.23–1.99) 24.29 ^a^ (22.31–24.30)	2.7 (2.3–3.0) 13.76 (12.74–14.78) 57.15 (57.14–57.16)	2.5 (2.1–2.8) 13.41 ^b^ (13.35–13.43) 19.27 ^b^ (19.26–19.28)	2.8 (2.5–3.3) 10.98 (8.95–11.99) 32.88 (32.86–32.89)

**Table 6 nanomaterials-08-01022-t006:** Predicted steady-state plasma concentration Css of the PF after the application of each formulation on 100 cm^2^ of skin. Therapeutic plasma concentration: 4.89 ± 1.29 µg/mL and 10.19 ± 2.43 µg/mL for young and elderly subjects, respectively. Data are represented as median (min-max).

Formulation	Young Subject Css (µg/mL) × 10^3^	Elderly Subject Css (µg/mL) × 10^3^
NLCs-F3	8.60 (7.30–11.80)	16.36 (13.75–22.25)
NLCs-F3-L	1.06 (0.99–1.14)	1.99 (1.85–2.15)
NLCs-F9	1.08 (1.07–1.09)	2.04 (2.03–2.05)
NLCs-F9-L	1.32 (1.21–1.62)	2.49 (2.28–3.06)

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
