# Peer review of "Development of Pranoprofen Loaded Nanostructured Lipid Carriers to Improve Its Release and Therapeutic Efficacy in Skin Inflammatory Disorders"

_nanomaterials, 2018, doi:10.3390/nano8121022_

Reviewer 1 Report

Consider breaking this into 3 manuscripts, or, if available, place some of the information into an appendix. The physical characterization of the nanoparticles would be a good candidate for an appendix. Also Tables 1, 2 & 3 could be appended.

line 35 "linoleic" --> linoleic line 36 "Interleukins" --> interleukins

line 53, References 3 & 4 support a trancellulat pathway through the stratum corneum; however, this view is not widely accepted. The paracellular route is much more widely accepted.

lines 73-74 "It has poor aqueous solubility as a function of the media pH [7,8]." Does the solubilityy vary with pH? Please explain.

section 2.2.4 Stability -- Check of the chemical stability of the permeabilizers should also be included in this. Some of the permeabilizers are readily oxidized. It does no good to have nanoparticles that are stable if the active components break down.

Model of mice ear inflammation induced with TPA Was this approved by the ethical committee?

Figure 8.Which bars are significantly different from which other bars?

Reference 1 As listed, this reference could not be found.The authors and year are incorrect. Am J Clin Dermatol 19:103-117 is actually Vaughn AR, Clark AK, Sivamani RK, Shi VY: Natural oils for skin-barrier repair:  Ancient compounds now backed by modern science. 2018

lines 321-322 "The skin was cleaned using a 0.05% solution of sodium laurylsulfate and washed in distilled water." How sure are the authors that this removes all of the applied formulation? This could be tested by cleaning and washing a control shortly after application of the formulation and then checking for residual PF.

Author Response

REVIEWER 1

Comments and Suggestions for Authors

Consider breaking this into 3 manuscripts, or, if available, place some of the information into an appendix. The physical characterization of the nanoparticles would be a good candidate for an appendix. Also Tables 1, 2 & 3 could be appended.

Tables 1, 2, 3 and 4 have been added as supplementary material.

line 35 "linoleic" --> linoleic line 36 "Interleukins" --> interleukins

Errors have been corrected.

line 53, References 3 & 4 support a trancellulat pathway through the stratum corneum; however, this view is not widely accepted. The paracellular route is much more widely accepted.

We share the reviewer’s concern and we have adapted the text accordingly. It now states “The lipid layers are anisotropic, meaning that their permeability is direction-dependent. This property greatly affects the diffusion characteristics in the skin, since diffusion through the SC is intercellular [3] [4].”

References 3 & 4 have been deleted and subsequently replaced with two new references

- RodriguesLeite-Silva V, Mandelli de Almeida M, Fradin A, Grice JE, Stephen M. Delivery of drugs applied topically to the skin. Expert Rev Dermatol 2012;7(4):383-397.

- Asbill CS, Michniak BB. Percutaneus penetration enhancers: local versus trasdermal activity. Pharm Sci Technolo Today. 2000;3(1):36-41.

lines 73-74 "It has poor aqueous solubility as a function of the media pH [7,8]." Does the solubilityy vary with pH?  Please explain

A solubility test was performed using an HPLC method under the conditions described in section 3.6.1. of this manuscript, with pranoprofen dissolved in distilled water pH 6.5, water milliQ pH 6.0 and Buffer phosphate Sorensen pH 7.4 1/18 M, obtaining 2.02 mg / mL, 2.16 mg / mL and 0.706 mg / mL respectively.

section 2.2.4 Stability -- Check of the chemical stability of the permeabilizers should also be included in this. Some of the permeabilizers are readily oxidized. It does no good to have nanoparticles that are stable if the active components break down.

When NLCs were prepared the linoleic acid was included in the lipid matrix. Therefore linoleic acid is inside nanoparticles and the compound is protected from any external agent that could cause its degradation.

Model of mice ear inflammation induced with TPA Was this approved by the ethical committee?

Indeed. The Ethics Committee approved the full protocol. Please see attached the EC approval form.

Figure 8. Which bars are significantly different from which other bars?

In Figure 8, the main objective is to compare the Blank samples (light grey bars) with NLCs-F3-L (dark grey bars) and NLCs-F9-L (black bars) in each dilution assayed independently, but not between dilutions. In all cases the statistical analysis (one way-ANOVA) between different groups was p>0.05 which means than there are no significative differences between the three groups for each dilution. For this reason, there are no asterisks of significance in this plot.

Reference 1 As listed, this reference could not be found.  The authors and year are incorrect. Am J Clin Dermatol 19:103-117 is actually Vaughn AR, Clark AK, Sivamani RK, Shi VY: Natural oils for skin-barrier repair:  Ancient compounds now backed by modern science. 2018

Erratum in reference 1 has been corrected.

lines 321-322 "The skin was cleaned using a 0.05% solution of sodium laurylsulfate and washed in distilled water." How sure are the authors that this removes all of the applied formulation? This could be tested by cleaning and washing a control shortly after application of the formulation and then checking for residual PF.

A new ex vivo human skin permeation assay of PF from NLCs F3 and NLCs F3-L formulations using skin samples from the same donor and performed under the conditions described in this manuscript and without sampling in the sampling times was performed. At the end of the study, the skin was used to determine the amount of drug retained and the skin was cleaned using a 0.05% solution of sodium laurylsulphate (n=3) or with a mixture of solvents Trascutol:ethanol:water(50:25:25) (n=3). The results obtained in this study are reported in the following table:

Solvent

NLC-F3

NLC-F3-L

0.05% Sodium   laurylsulfate

25.02 (21.54 – 27.13 )

55.24 (52.33 – 58.00 )

Trascutol:ethanol:water(50:25:25)

23.21 (20.09 – 28.22)

54.04 (51.60 – 57.56)

Taking into account the results obtained it is possible to conclude that the formulation is completely eliminated from the skin when it is cleaned with any of the solvents used, obtaining similar results.

Reviewer 2 Report

The subject matter of this paper deals with the development of nanostructured lipid carriers to improve the release and therapeutic efficacy of pranoprofen in skin inflammatory disorders. The manuscript itself can be considered relevant even if there are some critical concerns that should be addressed and clearly resolved:

1) Please read the paper carefully, for English language style and spelling, and make appropriate corrections and changes. In addition, please define abbreviations/acronyms at first use in text.

2) The authors report results on several formulations (NLCs-F3, NLCs-F9; NLCs-F3-L and NLCs-F9-L). Anyway, the compositions are not clearly reported.  The authors should explain and report them in the method section.

3) Entrapment efficiency is calculated by indirect method but this approach has many disadvantages (NLCs precipitation could be incomplete/ it does not take into account the adsorbed drug molecules/etc.). The authors should try to assess the EE% by a direct method.

4) All formulations have Zp values on average around -10 mV. On the basis of these values (table 3), in the Discussion Section (lines 760-763) the authors state that all the formulations had a net negative charge able to prevent aggregation. I am not convinced about it: generally, ZP value is considered satisfactory, under a stability point of view, when it is around ± 30.

5) Regarding TEM figures: the magnification should be added. The size of the particles in the TEM figures are different from Z-Ave (size) obtained by LS. The authors should discuss it.

The authors state that TEM images confirm the adequacy of the preparation method but I am not convinced about it. Please provide additional results of microscopic analysis.

6) lines 314-319: In the method the human skin permeation studies with penetration enhancers were poorly explained: the description of the employed procedure should be detailed. The term “promoter” should be replaced with permeation enhancer.

7) Line 699, please write the number of the Figure.

8) Lines 794-799 report a description of the MTT method instead of a discussion on the in vitro cytotoxicity studies. The authors should review this part of the manuscript.

9) The CSS values reported in Table 7 are higher than the therapeutic plasma concentration of PF. If so, in my opinion it is not clear the sentence of lines 616-617. In addition, the Css value obtained for the selected NLCs-F3 (line 645) formulation is higher than the therapeutic plasma concentration. Please discuss all these points.

10) Regarding in vivo studies: ten volunteers, six adult male Wistar CD-1 mice and three New 409 Zealand male albino rabbits were employed in the experiments. The authors should consider to expand the in vivo studies and provide additional comments on the statistical significance of their results.

11) The skin irritation testing is poorly described. I strongly recommend that authors try to show some data demonstrating they results.

Author Response

REVIEWER 2

Review

Comments and Suggestions for Authors

The subject matter of this paper deals with the development of nanostructured lipid carriers to improve the release and therapeutic efficacy of pranoprofen in skin inflammatory disorders. The manuscript itself can be considered relevant even if there are some critical concerns that should be addressed and clearly resolved:

1)Please read the paper carefully, for English language style and spelling, and make appropriate corrections and changes. In addition, please define abbreviations/acronyms at first use in text.

The paper has undergone an extensive review of English spelling and style and corrections have me made.

2)The authors report results on several formulations (NLCs-F3, NLCs-F9; NLCs-F3-L and NLCs-F9-L). Anyway, the compositions are not clearly reported.  The authors should explain and report them in the method section.

A new Table 1 has been added and the composition and Physicochemical characterization of the different formulations used are reported.

3)Entrapment efficiency is calculated by indirect method but this approach has many disadvantages (NLCs precipitation could be incomplete/ it does not take into account the adsorbed drug molecules/etc.). The authors should try to assess the EE% by a direct method.

In this investigation, the entrapment efficiency was determined by the indirect method, taking into consideration that the literature reports concordance between the results obtained by both the direct method and the indirect method in this particular setting.

- Beny Baby, Nagaraja Sree Harsha, Korlakunta Narasimha Jayaveera, Abin Abraham. Formulation and Evaluation of Levofloxacin Nanoparticles by Ionic Gelation Method. Research and Reviews: Journal of Pharmacy and Pharmaceutical Sciences. 2012;1(1):7-15

- Gonçalves L.M.D., Maestrelli F., Di Cesare Mannelli L., Ghelardini C., Almeida A.J., Mura P. Development of solid lipid nanoparticles as carriers for improving oral bioavailability of glibenclamide. European Journal of Pharmaceutics and Biopharmaceutics 2016;102: 41–50

- Khoshneviszadeh R, Sedigheh Fazly Bazzaz B, Reza Housaindokht M, Ebrahim-Habibi A, Rajabi O. A Comparison of Explanation Methods of Encapsulation Efficacy of Hydroquinone in a Liposomal System

Additionally, there is a great diversity of investigations in which the indirect method has been used to determine this property of the nanoparticles.

- Gonzalez-Mira E, Egea MA, Garcia ML, Souto EB. Design and ocular tolerance of flurbiprofen loaded ultrasound-engineered NLC. Colloids and surfaces B, Biointerfaces 2010;81:412-21.

- Sánchez-López E, Ettcheto M, Egea MA, Espina M, Calpena AC, Folch J, et al. New potential strategies for Alzheimer's disease prevention: pegylated biodegradable dexibuprofen nanospheres administration to APPswe/PS1dE9. Nanomedicine: Nanotechnology, Biology and Medicine 2017;13:1171-82

4)All formulations have Zp values on average around -10 mV. On the basis of these values (table 3), in the Discussion Section (lines 760-763) the authors state that all the formulations had a net negative charge able to prevent aggregation. I am not convinced about it: generally, ZP value is considered satisfactory, under a stability point of view, when it is around ± 30.

We share the reviewer’s concern that ZP values of charge are not high enough to prevent aggregation per se for electrostatic attraction. However, it should be taken into account that the surfactant used in the preparation of the nanoparticles is Tween 80, which is a non-ionic surfactant that it is stabilized by steric effect. Literature supports that the use of this surfactant is sufficient to prevent aggregation and achieve stability under these circumstances. 

E. Gonzalez Mira, M.a. Egea M.L.Egea E.B. Souto Design and ocular tolerance of flurbiprofen loaded ultrasound-engineered NLC E. Gonzalez Mira, M.a. Egea Colloids and Surfaces B:Biointerfaces

The data obtained in the short and long term stability studies confirms that the formulations are stable

5)Regarding TEM figures: the magnification should be added. The size of the particles in the TEM figures are different from Z-Ave (size) obtained by LS. The authors should discuss it.

The authors state that TEM images confirm the adequacy of the preparation method but I am not convinced about it. Please provide additional results of microscopic analysis.

As it can be seen in the TEM image, the sizes of the nanoparticles are lower than those obtained in the physicochemical characterization DLS. Scale (in micrometers) is stated in the bottom right of the image. Magnification has been added (60,000 x magnification). 

The increase in the particle size could be attributed due to that the nanoparticles in suspension were highly hydrated and the mean particle size measured by PCS was "hydrated diameter", which is often larger than their real size.

Song X. Zhao Y. Hou S. Xu F. Zhao R. He J. Cai Z. Li Y. Chen Q 2008. Dual agents loaded PLGA nanoparticles: Systematic study of particle size and drug entrapment efficiency. European Journal of Pharmaceutics and Biopharmaceutics 69:445-453.7

6)lines 314-319: In the method the human skin permeation studies with penetration enhancers were poorly explained: the description of the employed procedure should be detailed. The term “promoter” should be replaced with permeation enhancer.

The text has been appropriately updated, with the following addition “For the selection of permeation enhancers (pyrrolidone, decanol, octanoic acid, nonane, menthone, squalene, linoleic acid and cineol) skin permeation studies were made using Franz diffusion cells (FDC 400; Crown Glass, Somerville, NJ, USA) using human skin membranes and PBS at pH 7.4 as a medium with a skin area available for permeation of 0.64 cm2. Samples of 1mL of PF diluting in PBS pH 7.4 (1mg/mL) with 5% v/v of each permeation enhancer were placed in the donor compartment. Samples (300 μL) were withdrawn from the receptor compartment at fixed times and replaced by an equivalent volume of PBS pH 7.4 solution at the same temperature. At the end of the study, the skin was used to determine the amount of drug retained. The skin was cleaned using a 0.05 % solution of sodium laurylsulphate and washed in distilled water. The diffusional area of the skin in direct contact with the formulation was isolated, weighed and treated with methanol: water (50:50, v:v) during 20 minutes under sonication in an ultrasound bath..”

The term “promoter” has been replaced by “permeation enhancer”.

7)Line 699, please write the number of the Figure.

The number of figure has been added.

8)Lines 794-799 report a description of the MTT method instead of a discussion on the in vitro cytotoxicity studies. The authors should review this part of the manuscript.

This part of the manuscript has been modified by: “HaCaT cell line is a common in vitro skin model that has been widely used to test the cell cytotoxicity of different nanoparticles [43, 44]. In this study, cells were incubated for 24 h with different concentrations of optimized NLCs and Blank-NLCs as reference to compare. Results from figure 7 shows in general that Blank-NLCs (not loaded with PF) has the same cytotoxicity profile than those that carry the anti-inflammatory compound, which would be indicating that the cellular damage is not due to the PF concentration if not the NLCs composition. This result agrees with previous studies performed by Graham et al. where they assayed higher concentrations of nanoparticles carrying similar anti-inflammatory components using the same HaCaT model [45].

On the other hand, when different dilutions of NLCs-F3-L, NLCs-F9-L and Blank-NLCs were tested, it was observed, for all nanoparticles types, that samples more diluted than 1/50 barely affected cell viability, with a 95% of cell survival. However, when HaCaT were incubated with nanoparticles samples more concentrated than 1/20, it affected negatively to cell viability in a concentration-dependent manner. This result confirms that maybe one or some components of the NLCs at high concentrations, are the responsible for the cell death. Thus, only PF concentrations from 0.3 to 12 µg/mL for NLCs-F3-L, and from 0.2 to 8 µg/mL for NLCs-F9-L should be used as suitable for topical use, due to the absence of toxicity.”.

[1]. Eupafolin nanoparticles protect HaCaT keratinocytes from particulate matter-infuced inflammation and oxidative stress. Zih-Chan Lin, Chiang-Wen Lee, Ming-Homg Tsai, Homg-Huey Ko, Jia-You Fang, Yao-Chang Chiang, Chan-Jung Liang, Lee-Fen Hsu, Stephen Chu-Sung Hu, Feng-Lin Yen. Int J Nanomedicine. 2016; 11: 3907-3926. doi:10.2147/IJN.S109062.

[2]. Crosera M., Prodi, A., Mauro, M., Pelin, M., Florio, C., Bellomo, F., G, Apostoli, P., De Palma, G., Bovenzi, M., Cmapanini, M., Filon F. L. Titanium dioxide nanoparticle permetration into the skin and effects of HaCaT cells. Int J Env Re and Public Health. 2015; 12(8), 9282-97. Doi:10.3390/ijerph120809282.

[3]. S. Graham, R. Philip, M. Zahid, N. Bano, Q. Iqbal, F. Mahboob, X. Chen, L. Shang. Ibuprofen nanoparticles and its cytotoxicity on A549 and HaCaT cell lines. Proc Physiol Soc 37. PAC 138.

9)The CSS values reported in Table 7 are higher than the therapeutic plasma concentration of PF. If so, in my opinion it is not clear the sentence of lines 616-617. In addition, the Css value obtained for the selected NLCs-F3 (line 645) formulation is higher than the therapeutic plasma concentration. Please discuss all these points.

All Css values reported for the permeation enhancer are lower than therapeutic plasma concentration (4.89 ± 1.29 µg/mL and 10.19 ± 2.43 µg/mL for young and elderly subjects respectevely). Css x 103 =  value (µg/mL).  For example value obtained for Pyrrolidine Css x 103 = 96.41 µg/mL; Css (µg/mL) = 96.41 / 103 ;  Css =94.41 x 10-3  µg/mL  or 0.09641 µg/mL which is lower than therapeutic plasma concentration.

Css value reported for NLCs-F3 is 16.36 x 10-3 µg/mL or 0.01636 µg/mL, which is lower than therapeutic plasma concentration

10)Regarding in vivo studies: ten volunteers, six adult male Wistar CD-1 mice and three New 409 Zealand male albino rabbits were employed in the experiments. The authors should consider to expand the in vivo studies and provide additional comments on the statistical significance of their results.

All the in vivo assays performed in this study have been carried out in accordance with a protocol that has been approved by the corresponding ethical committees that have evaluated the research. The certificates issued by the Ethical Committee for each study has been sent to the editor of the journal.

The number of animals employed in the in vivo studies was adapted to meet the requisites issued by the Ethical Committee under the principle of “Reduce, Reuse, Recycle” for the use of animals in research. An increase of the sample would entail more precise information, although we consider that the current numbers allow to obtain solid conclusions and are similar to comparable studies reported in the literatura.

Regarding the number of volunteers; the null hypothesis was that there is no difference in stratum corneum hydration with the use of selected NLCs. Using a two-sided level of significance of α=0.05, this assumption resulted in 10 volunteers needed for an 80% power to reject the null hypothesis. Skin occlusion can increase the stratum corneum hydration and figure 12 showed increase in hydration values in all of the selected formulations. An statistically significant increase was identified one hour after application. 

11) The skin irritation testing is poorly described. I strongly recommend that authors try to show some data demonstrating they results.

This part of the manuscript has been modified by “Figure Supplementary 2 shows in vivo Draize skin test of the selected NLCs that was assessed in male albino rabbits. No erythema, nor edema were found after 24, 48 and 72 h of exposure, and no signs of skin irritancy were detected, since the individual primary irritancy index determined for these formulations were less than 0.5.”

It has been added in Supplementary material Figure Supplementary 2:

Reviewer 3 Report

General comments

In this manuscript, the authors describe the preparation and characterization of pranoprofen (PA) loaded nanostructured lipid carriers (NLCs). The authors performed several experiments to evaluate the anti-inflammatory activity and safety of the investigated PA loaded NLCs. The topic of this manuscript is interesting but the experimental protocol is not properly described and the results are not clearly presented.

Specific comments.

Line 24. The meaning of the sentence “A study to optimize and characterize Pranoprofen (PF)-loaded nanostructured lipid carriers (NLCs), has been carried out by high-pressure homogenization method” is unclear. High-pressure homogenization is a method to prepare NLCs and it cannot be used to optimize or characterize NLCs. Please, rephrase this sentence.

Line 53. The statement “diffusion through the SC is transcellular” is in contrast with the evidence reported in literature as the intercellular route is regarded the main pathway for drug diffusion through the SC (Rodrigues Leite-Silva et al., Expert Review of Dermatology, 7:4, 383-397, DOI: 10.1586/edm.12.32). Please, modify this statement in accordance with the literature.

Line 89-90. Linolelic should read linolenic.

Line 128. The meaning of the abbreviation MTT should be explained. Please, specify the meaning of all abbreviations and acronyms at their first use in the text.

Line 139. The composition of the prepared NLCs is unclear. The authors should provide a Table reporting the composition of all prepared NLCs.

The authors should report Table 3 in the results section.

Line 202. The term “nanoemulsions” is incorrect. Please, modify.

Line 220. The method to evaluate sample spreadability is not well described. The authors should specify which “increasing standard weights” they used. In addition, they should report how they validated this method or suitable references.

Line 257. The authors performed in vitro release studies using Franz-type diffusion cells. The authors should report all features of these diffusion cells (surface area exposed to drug release/permeation, receptor and volume of samples applied in the donor compartment). The authors used a phosphate buffer pH 7.4 as receptor medium. In the introduction, the authors reported that PA is poorly water-soluble. What was PA solubility in the phosphate buffer used as receptor medium? How did the authors verify that sink conditions were ensured using this receptor medium?

Line 287. The composition of NLCs-F3-L and NLCs-F9-L is not reported in the text. What is the meaning of sentence “using a stock pranoprofen concentration of 0.712 mg/ml and 0.470 287 mg/ml respectively”? Why did the authors use a stock solution of PA?

Line 346. The histological analysis is not well detailed. The authors should report a better description of this analysis adding the method of skin treatment, the number of treated animals, the approval of these experiments by an ethical committee, etc. 

Line 365. It is unclear how the authors performed in vivo studies on human volunteers. According to the results of viscosity determinations, these samples were liquid. How did the authors apply the samples? What amount of samples did they apply?

Line 394. How did the authors treat mice`s ear tissue with the samples under investigation (amount of sample, application surface area, in vitro or in vivo)? What was the control in these experiments?

Line 451. The unit of measure is missing.

The size and PI of NLCs showed in Fig. 1 A seem different from those obtained by PCS. The authors should explain this discrepancy.

Line 485. The results of spreadability determination are unclear. What are Y max and K parameters? The authors did not explain these parameters in the materials and methods section making the results of such determinations difficult to understand.

In Fig. 4, the code of each panel is missing.

Line 512. As the authors performed long-term stability studies, short-term stability studies are pointless and should be omitted.

Line 555. The authors reported that the maximum concentration of drug release (Q∞) from NLCs-F3 and NLCs-F3-L was higher in comparison with NLCs-F9 and NLCs-F9-L. Is this difference statistically significant? Please, add comments on the statistical significance.

Line 575. The meaning of the sentence “However, above 1/50 dilution, which correspond to 12 (NLCs-F3-L) and  6 (NLCs-F9-L) μg/ml of PF.” is unclear. Please, correct.

Line 663. The legend of Figure 10 is reported twice. Please, correct.

Figure 11 is not cited in the text.

Line 699. Please, add the figure number.

Line 823. Occlusive vehicles such as NLCs are supposed to increase skin hydration but the authors did not observe any increase of skin hydration after topical application of the investigated NLCs. The authors should provide an explanation of the lack of hydrating effect of these NLCs.

English must be carefully revised.

Author Response

REVIEWER 3

Open Review

Comments and Suggestions for Authors

General comments

In this manuscript, the authors describe the preparation and characterization of pranoprofen (PA) loaded nanostructured lipid carriers (NLCs). The authors performed several experiments to evaluate the anti-inflammatory activity and safety of the investigated PA loaded NLCs. The topic of this manuscript is interesting but the experimental protocol is not properly described and the results are not clearly presented.

Specific comments.

Line 24. The meaning of the sentence “A study to optimize and characterize Pranoprofen (PF)-loaded nanostructured lipid carriers (NLCs), has been carried out by high-pressure homogenization method” is unclear. High-pressure homogenization is a method to prepare NLCs and it cannot be used to optimize or characterize NLCs. Please, rephrase this sentence.

We agree with the reviewer and the sentence has been modified in the manuscript by: “Pranoprofen (PF)-loaded nanostructured lipid carriers (NLCs), prepared by high-pressure homogenization method, has been optimized and characterized, to improve the biopharmaceutical profile of this drug”.

Line 53. The statement “diffusion through the SC is transcellular” is in contrast with the evidence reported in literature as the intercellular route is regarded the main pathway for drug diffusion through the SC (Rodrigues Leite-Silva et al., Expert Review of Dermatology, 7:4, 383-397, DOI: 10.1586/edm.12.32). Please, modify this statement in accordance with the literature.

We share the reviewer’s concern and we have adapted the text accordingly. It now states “The lipid layers are anisotropic, meaning that their permeability is direction-dependent. This property greatly affects the diffusion characteristics in the skin, since diffusion through the SC is intercellular [3] [4].” 

References 3 & 4 have been deleted and subsequently replaced with two new references

- RodriguesLeite-Silva V, Mandelli de Almeida M, Fradin A, Grice JE, Stephen M. Delivery of drugs applied topically to the skin. Expert Rev Dermatol 2012;7(4):383-397.

- Asbill CS, Michniak BB. Percutaneus penetration enhancers: local versus trasdermal activity. Pharm Sci Technolo Today. 2000;3(1):36-41.

Line 89-90. Linolelic should read linolenic.

This mistake has been corrected.

Line 128. The meaning of the abbreviation MTT should be explained. Please, specify the meaning of all abbreviations and acronyms at their first use in the text.

MTT abbrevation meaning has been added: 3-(4,5-dimethylthiazol-2-yl)-2,5-diphenyltetrazolium bromide. Abbreviations and acronyms at their first use in the text have been reviewed to ensure that they are properly introduced.

Line 139. The composition of the prepared NLCs is unclear. The authors should provide a Table reporting the composition of all prepared NLCs.

In response to this comment together with other reviewer’s four tables have been moved to supplementary material and a new Table 1 has been added with the composition of the selected NLCs. The following tables are now located in the supplementary material section: Table S1 (Homogenization assays to select factorial design conditions), Table S2 (Factors and their corresponding levels in experimental design), Table S3 (Design of experiments of PF loaded Lipid NLCs) and Table S4 (Particle size of optimized formulations determined by LD).

The authors should report Table 3 in the results section.´

Following additional requests, previos Table 3 has been moved to Supplementary materials. However, in the current version of the manuscript, a new Table 1 has been included in the Results section with the physicochemical composition and characterization of the selected NLCs. 

Line 202. The term “nanoemulsions” is incorrect. Please, modify.

The term “nanoemulsions” has been modified by “NLCs”.

Line 220. The method to evaluate sample spreadability is not well described. The authors should specify which “increasing standard weights” they used. In addition, they should report how they validated this method or suitable references.

We share the reviewer’s concern and we have adapted the text accordingly. It now states  “A weight of 0.05 g of formulation was placed within a circle of 10 cm diameter pre-marked on a glass plate, over which second glass plate was placed, as centered as possible. Increasing standard weights pieces (5, 10, 15, 25 and 50 g) were replaced and allowed to rest on the upper glass plate for 1 minute. The increase in the diameter due to formulation spreading was noted. Each formulation was tested in triplicate at room temperature.”

References to valídate this method: Campaña-Seoane M, Peleitero A, Laguna R., Otero-Espinar FJ. Bioadhesive emulsions for control release of Progesterone resistant to vaginal fluids clearance. Internacional Journal of Pharmaceutics. 2014;477:495-505. 

The following images show the instrument that we use:

Line 257. The authors performed in vitro release studies using Franz-type diffusion cells. The authors should report all features of these diffusion cells (surface area exposed to drug release/permeation, receptor and volume of samples applied in the donor compartment). The authors used a phosphate buffer pH 7.4 as receptor medium. In the introduction, the authors reported that PA is poorly water-soluble. What was PA solubility in the phosphate buffer used as receptor medium? How did the authors verify that sink conditions were ensured using this receptor medium?

The manuscript has been modified by: “Amber glass vertical Franz diffusion cells (FDC 400; Crown Glass, Somerville, NJ, USA) with dialysis membranes (Dialysis Tubing Visking, Medicell International Ltd., London, UK) was used to address the PF release studies. The membrane was previously hydrated in methanol-water (7:3; v/v) for 24 hours before being mounted in the Franz diffusion cell. Phosphate buffer saline (PBS) at pH 7.4 kept at 32 ± 0.5 °C was used as receptor which was under continuous stirring at 700 rpm assuring sink conditions. Samples of 1440 µl of NLCs-F3 and NLCs-F3-L and 2160 µl of NLCs-F3 and NLCs-F3-L formulations were placed in the donor compartment in direct contact with the membrane (0.64 cm2), 300 µl of sample were collected with a syringe from the receptor compartment at predefined times and the volume withdraw was replaced by an equivalent volume of fresh PBS pH 7.4 at the same temperature [27]. Samples were analyzed by RP-HPLC as described previously for EE.

The amount of PF release was adjusted to four kinetic models: zero order, first order, Korsmeyer-Peppas and Weibull functions by nonlinear least-squares regression using the WinNonLin®software (WinNonlin®Professional edition version 3.3; Pharsight Corporation, Sunnyvale, CA, USA) and Graphpad prism version 6 Demo. The model fitting appropriateness was determined by calculation of the coefficient of determination (r2), and a discrimination models' parameter, the Akaike’s information criterion (AIC), that is a measure of the best fit based on maximum likelihood and was calculated by the equation:

AIC = n x ln (WSSR) + 2 x p          (2)

where n is the number of dissolution data points, p is the number of the parameters of the model and WSSR is the weighed sum of square of residues [22].”

On the other hand, PBS pH 7.4 was used as a receptor medium to simulate conditions similar to physiological pH, representing the pH of blood.

The dose applied in the donor compartment verify that sink conditions were used in release studies.

Line 287. The composition of NLCs-F3-L and NLCs-F9-L is not reported in the text. What is the meaning of sentence “using a stock pranoprofen concentration of 0.712 mg/ml and 0.470 287 mg/ml respectively”? Why did the authors use a stock solution of PA?

A new Table 1 has been added and the composition and Physicochemical characterization of the different formulations used are reported.

NLCs-F3-L formulation carry in their structures a concentration of 0.712 mg/mL of PF, but the formulation NLCs-F9-L was elaborated with 0.470 mg/ml of PF. These first NLCs solutions were referred in the manuscript as “stock” because were used to make different dilutions to assay the cellular cytotoxicity. Thus, dilutions were made using fresh DMEM medium, which means that it was assayed different NLCs and PF concentration.

As we cannot have a specific data of the number of NLCs that we have in a ml of solution, numbers were referred as PF concentration NLCs carry once they were generated.

Line 346. The histological analysis is not well detailed. The authors should report a better description of this analysis adding the method of skin treatment, the number of treated animals, the approval of these experiments by an ethical committee, etc. 

The following paragraph has been added: “Ears of the mice were subjected to the topical application of Arachidonic acid (AA, 5%) for 5 minutes as a model of skin inflammation. Then PF (0.75 mg/mL), NLCs-F3-L or NLCs-F9-L formulations were applied for 1 hour to study the anti-inflammatory effect. Ears of non-treated animals were used as control condition. After that mice were euthanized and ears were immediately excised, rinsed with PBS pH 7.4 and set overnight in 4% buffered formaldehyde at room temperature and then samples were embedded in paraffin wax. Transversal sections (5 µm) were stained with hematoxylin and eosin, and finally viewed on blind coded samples under a light microscope (Olympus BX41 and camera Olympus XC50) for the evaluation of the ear inflammation.”

All the in vivo assays performed in this study have been carried out in accordance with a protocol that has been approved by the corresponding ethical committees that have evaluated the research. The certificates issued by the Ethical Committee for each study has been sent to the editor of the journal.

Line 365. It is unclear how the authors performed in vivo studies on human volunteers. According to the results of viscosity determinations, these samples were liquid. How did the authors apply the samples? What amount of samples did they apply?

It has been added “Circles of 7 cm in diameter were made with a special marker of the skin. Samples of 50 µL equivalent to 37.5 µg of NLCs-F3 or NLCs-F3-L, and 25 µg of NLCs-F9 or NLCs-F9-L were deposited with an automatic micropipette in the middle of the circle by means of a soft circular movement with the thumb to facilitate the distribution of the formulation and with a chronometer was measured (2 turns / second). A total of 20 circles in a clockwise direction were performed. Measurements were performed before applying selected PF-NLCs and 5 and 60 minutes after application.”

Reference: Torres E, Suner Carbó J, Aróztegui M, Halbaut L, Barbé C. Propuesta de protocolo de anàlisis sensorial en productos semisólidos:  cremas i geles. NCP Documenta (Noticia de Cosmética y Perfumería). 2001; 260: 5-11.

Line 394. How did the authors treat mice`s ear tissue with the samples under investigation (amount of sample, application surface area, in vitro or in vivo)? What was the control in these experiments?

They were In vivo  assays and we analyzed mice dorsal ear´s tissue. The surface area was around 0.8 cm2. The amount of sample was 0.5 mL and was applied with a cotton swab. 

Two controls were established for comparison purposes: a negative control animal that did not receive any intervention, and a sham (+ control) for whom inflammation was induced with arachidonic acid, but did not receive anti-inflammatory treatment.We have included these informations in the methods section.

Line 451. The unit of measure is missing.

Unit of meassure has been added.

The size and PI of NLCs showed in Fig. 1 A seem different from those obtained by PCS. The authors should explain this discrepancy.

As it can be seen in the TEM image, the sizes of the nanoparticles are lower than those obtained in the physicochemical characterization DLS. Scale (in micrometers) is stated in the bottom right of the image. Magnification has been added (60,000 x magnification). 

The increase in the particle size could be attributed due to that the nanoparticles in suspension were highly hydrated and the mean particle size measured by PCS was "hydrated diameter", which is often larger than their real size.

Song X. Zhao Y. Hou S. Xu F. Zhao R. He J. Cai Z. Li Y. Chen Q 2008. Dual agents loaded PLGA nanoparticles: Systematic study of particle size and drug entrapment efficiency. European Journal of Pharmaceutics and Biopharmaceutics 69:445-453.7

Line 485. The results of spreadability determination are unclear. What are Y max and K parameters? The authors did not explain these parameters in the materials and methods section making the results of such determinations difficult to understand.

Thanks for your comment, the equations of the graphs correspond to first order kinetic model, being the best fit model these formulations. We have included these informations in the methods and results sections.

In Fig. 4, the code of each panel is missing.

Codes of panel in Figure 4 have been added.

Line 512. As the authors performed long-term stability studies, short-term stability studies are pointless and should be omitted.

Figure of short-term stability has been added as supplementary material

Line 555. The authors reported that the maximum concentration of drug release (Q∞) from NLCs-F3 and NLCs-F3-L was higher in comparison with NLCs-F9 and NLCs-F9-L. Is this difference statistically significant? Please, add comments on the statistical significance.

We share the reviewer’s concern, the maximum concentration of release drug (Q∞) from NLCs-F3 and NLCs-F3-L was higher in comparison with NLCs-F9 and NLCs-F9-L, but this difference is not statistically significant, and we have adapted the text accordingly. It now states “As it can be observed, the release profile of PF from NLCs-F9 and NLCs-F9-L showed faster release than from NLCs-F3 and NLCs-F3-L, however the maximum concentration of release drug (Q∞) from NLCs-F3 and NLCs-F3-L was higher in comparison with NLCs-F9 and NLCs-F9-L, but this difference is not statistically significant.”

Line 575. The meaning of the sentence “However, above 1/50 dilution, which correspond to 12 (NLCs-F3-L) and  6 (NLCs-F9-L) μg/ml of PF.” is unclear. Please, correct.

We agree with the reviewer and the sentence has been modified in the manuscript by “However, concentrations of pranaprofen (μg/ml) below 12 for NLCs-F3-L and 8 for NLCs-F9-L barely affected cell viability (98-100%)”

Line 663. The legend of Figure 10 is reported twice. Please, correct.

Error has been corrected.

Figure 11 is not cited in the text.

Figure 11 has been cited.

Line 699. Please, add the figure number.

Number of figure has been added.

Line 823. Occlusive vehicles such as NLCs are supposed to increase skin hydration but the authors did not observe any increase of skin hydration after topical application of the investigated NLCs. The authors should provide an explanation of the lack of hydrating effect of these NLCs.

We share the reviewer’s concern, skin occlusion can increase the stratum corneum hydration and figure 12 showed increase in hydration values in all of the selected formulations, although it was obtained only one statistically significant increase one hour after application. If the experiment would spent more time, it would be found more increse values.

English must be carefully revised.

The paper has undergone an extensive review of English spelling and style and corrections have me made.

Round 2

Reviewer 1 Report

OK

Reviewer 2 Report

I revised the new version of the manuscript and I believe It has been improved and now warrants publication in Nanomaterials.  

Reviewer 3 Report

The authors revised the manuscript properly.